# Sex-stratified genome-wide association meta-analysis of major depressive disorder

Jodi T. Thomas [1,2] ✉, Jackson G. Thorp [1,2], Floris Huider [3,4,5], Poppy Z. Grimes [6], Rujia Wang [7,8], Pierre Youssef [1], Jonathan R. I. Coleman [7,8], Enda M. Byrne [9], Mark Adams [6], BIONIC consortium*, The GLAD Study*, Sarah E. Medland [1], Ian B. Hickie [10], Catherine M. Olsen [11], David C. Whiteman [11], Heather C. Whalley [6], Brenda W.J.H. Penninx [5,12], Hanna M. van Loo [13], Eske M. Derks [1], Thalia C. Eley [7,8], Gerome Breen [7,8], Dorret I. Boomsma [4,5], Naomi R. Wray [14,15], Nicholas G. Martin [1] & Brittany L. Mitchell [1,2,16] ✉

There are striking sex differences in the prevalence and symptomology of Major Depressive Disorder. Here, we conduct the largest sex-stratified genome wide association and genotype-by-sex interaction meta-analyses of Major Depressive Disorder to date (Females: 130,471 cases, 159,521 controls. Males: 64,805 cases, 132,185 controls). We identify 16 and eight independent genome-wide significant variants in females and males, respectively, including one novel variant on the X chromosome. Major Depressive Disorder in females and males shows substantial genetic overlap with a large proportion of variants displaying similar effect sizes across sexes. However, we also provide evidence for a higher burden of genetic risk in females which could be due to female-specific variants. Additionally, sex-specific pleiotropic effects may contribute to the higher prevalence of metabolic symptoms in females with Major Depressive Disorder. These findings underscore the importance of considering sex-specific genetic architectures in the study of health conditions, including Major Depressive Disorder, paving the way for more targeted treatment strategies.

Major Depressive Disorder (MDD) exhibits striking sex differences in both prevalence and clinical presentation. Throughout this manuscript, the term 'sex' refers to differences in biological characteristics between females and males. Globally, females are nearly twice as likely as males to experience MDD, with this disparity emerging around puberty and persisting into adulthood[1,2]. This prevalence difference persists across a variety of forms of diagnosis, as well as across cultures and geographic borders[1,3]. Beyond variation in prevalence, sex differences extend to symptomatology. Females tend to have a higher prevalence of atypical depression, characterised by symptoms such as weight gain, hypersomnia, and increased appetite, as well as immuno-metabolic depression, which is defined by immune-inflammatory pathophysiology and metabolic dysregulation and often overlaps symptomatically with atypical depression. In contrast, males with MDD more frequently exhibit anger, aggression, risk-taking behaviours, and substance use, with higher rates of comorbid substance use disorders[4,5]. These differences suggest potential underlying biological and psychosocial mechanisms that contribute to MDD heterogeneity across sexes[6].

Multiple explanations have been proposed for this MDD heterogeneity across sexes, spanning behavioural, environmental, and biological domains. One potential explanation is variation in help-seeking and symptom reporting, as males are generally less likely to seek professional help or disclose symptoms, leading to under-diagnosis[7].

Environmental exposures also vary by sex; for example, females are more frequently exposed to sexual abuse and other forms of interpersonal violence, and experience structural forms of discrimination such as the gender wage gap, which may contribute to the higher prevalence of MDD in females[8,9]. Additionally, biological mechanisms may underlie these differences, with research pointing to the roles of the immune system, neuroanatomy, neuroplasticity, stress and the hypothalamic-pituitary-adrenal (HPA) axis, and hormonal influences such as sex hormones and the hypothalamic-pituitary-gonadal (HPG) axis[10–14]. Together, these factors highlight the complexity of MDD heterogeneity across sexes and underscore the need for a multifaceted approach to understanding its underlying mechanisms.

Despite these numerous potential explanations, there has been limited replicated research clarifying the true aetiology of these sex differences in MDD. A key component of the biological mechanisms underlying these disparities could be differences in genetics. Research suggests that sex differences in human complex traits, including MDD, may arise from both sex-dependent and sex-specific genetic effects[15–18]. Sex-dependent effects refer to genetic variants that affect both sexes, but with effect sizes that differ in magnitude or direction between females and males. Sex-specific effects suggest that different genetic variants may contribute to MDD in males and females. Moreover, genetic variants on the X chromosome may play a crucial role in these sex differences, highlighting the need to consider sex chromosomes in genetic studies[19,20].

There is mixed evidence for the role of genetics in the sex differences of MDD. A meta-analysis of five twin studies found no evidence for sex differences in the heritability of MDD[21]. However, other twin studies have shown higher heritability of MDD in females (40–51%) compared to males (29–41%), along with a genetic correlation between sexes that is significantly less than one[22–24]. More recently, genome-wide association studies (GWAS) have also provided mixed evidence for the existence of sex-dependent and sex-specific genetic effects contributing to these differences. For instance, a study using data from the UK Biobank reported a SNP-based genetic correlation between broad depression in males and females that was significantly less than one ($r_g = 0.91$, standard error (SE) not reported)[17], however this was not the case in a sex-stratified meta-analysis of MDD conducted by the Psychiatric Genomics Consortium ($r_g = 1.01$, SE = 0.2)[16]. In this same study, SNP-based heritability was estimated to be significantly higher in females than in males[16]. Furthermore, a cross-disorder (schizophrenia, bipolar disorder, MDD) genotype-by-sex interaction analysis identified a locus with opposite effects in females and males, further pointing to the role of sex-specific genetic factors[16]. These mixed results may stem from inconsistencies in cohort methodologies including case ascertainment[25]. Research has shown that genetic correlations can vary substantially across GWAS depending on the stringency and specificity of MDD phenotype definitions[26]. These findings underscore the need for consistent phenotyping approaches that align closely with diagnostic criteria to enhance the accuracy and comparability of genetic studies of MDD.

In this study we aimed to investigate whether sex differences in MDD can be explained, at least in part, by genetic effects. We conducted the largest sex-stratified GWAS and genome-wide genotype-by-sex interaction meta-analyses of MDD to date, using MDD cases primarily based on DSM (Diagnostic and Statistical Manual of Mental Disorders) criteria. Specifically, we examined whether genetic variants contribute to these differences through sex-dependent effects, sex-specific effects, or variants on the X chromosome. Additionally, we explored whether sex-specific pleiotropic effects between MDD and other phenotypes might help explain the observed differences in phenotypic presentation across sexes. These insights into sex-specific genetic mechanisms not only deepen our understanding of the aetiology of MDD but may also inform the development of novel therapeutics that are tailored to sex-specific genetic risk profiles,

ultimately contributing to more targeted and effective precision medicine strategies for MDD.

## Results

### Sex-stratified GWAS

We conducted genome-wide association studies (GWAS) of Major Depressive Disorder (MDD) in five new cohorts. These were meta-analysed with previously published GWAS meta-analysis summary statistics from Blokland et al.[16] for each sex separately. Sex was defined by chromosomal composition, with XX individuals as female and XY as male. The final sample size was 130,471 cases and 159,521 controls in females, and 64,805 cases and 132,185 controls in males (Supplementary Data 1). Our sex-stratified GWAS analyses identified 16 and eight independent genome-wide significant SNPs in females and males, respectively (Fig. 1). In males, one novel SNP was identified on the X chromosome (rs5971319), which has not previously been associated with any depression phenotypes (Supplementary Data 2–3).

Our sex-stratified results showed high genetic correlation with the largest GWAS sex-combined meta-analysis of MDD[27] (Female-both sexes $r_g = 0.98 \pm 0.01$ ($r_g \pm$ standard error)), Male-both sexes $r_g = 0.92 \pm 0.02$) (Supplementary Fig. 1, Supplementary Data 4-5), and little evidence for residual population stratification (Supplementary Fig. 2). We tested the independent genome-wide significant SNPs (16 in females and eight in males) in our replication cohort, Generation Scotland. The number of SNPs with a concordant effect size direction was not significantly greater than expected by chance in both females (concordance = 69% [95% CI: 45–100%], H0: concordance = 0.5 [$p = 0.11$]) and males (concordance = 71% [95% CI: 34–100%], H0: concordance = 0.5 [$p = 0.23$]) (Supplementary Data 6, 7). However, the replication cohort is small and the 95% confidence intervals are large suggesting the power of this replication is low.

### Genetic architecture of MDD in females and males

We investigated whether there was evidence for a sex difference in the genetic architecture of MDD by estimating sex-specific autosomal SNP-based heritability ($h^2_{SNP}$), polygenicity ($\pi$) and the selection parameter ($S$) using our sex-stratified meta-analysis results in SBayesS. $h^2_{SNP}$ was converted to the liability scale using a lifetime population risk of 0.2 in females and 0.1 in males. We found very strong evidence (100% posterior probability (PP)) that $h^2_{SNP}$ is higher in females than males ($h^2_{SNP}$ female = 11.3% [95% highest posterior density interval (HPDI): 10.7 – 11.9%]; $h^2_{SNP}$ male = 9.2% [8.4–9.9%]]) (Fig. 2a). This suggests that the amount of variation in MDD risk explained by SNPs is higher in females. There was very strong evidence (100% PP) that MDD polygenicity is higher in females ($\pi = 0.02$ [0.015–0.024]) than males ($\pi = 0.013$ [0.009–0.017]), suggesting more SNPs contribute to MDD risk in females (Fig. 2b). Univariate MiXeR[28] estimated 13,244 (SE = 1120) causal variants explain 90% of MDD $h^2_{SNP}$ in females versus 7111 (SE = 701) in males (Supplementary Data 8), further supporting higher polygenicity in females. Lastly, there was moderate evidence (79% PP) that the selection parameter was lower in males ($S = -0.14$ [−0.33–0.11]) than in females ($S = -0.05$ [−0.17–0.08]) (Fig. 2c). This suggests that the negative relationship between effect size and minor allele frequency, an indicator of negative natural selection, could be stronger in males than females.

We conducted a range of sensitivity analyses. For $h^2_{SNP}$, we used LDSC as an alternative method and explored the impact of lifetime population prevalence and unscreened controls (given evidence that MDD could be under-diagnosed in males)[29]. For $h^2_{SNP}$, $\pi$ and $S$, we 1) assessed the effect of differential power between the female and male GWAS by repeating analyses using equivalent sample sizes in females and males, and 2) accounted for across-cohort heterogeneity[25]. Overall, our sensitivity analyses showed the same pattern of sex differences with a higher $h^2_{SNP}$ and polygenicity in females compared to males, however sex differences in the selection parameter did not remain in

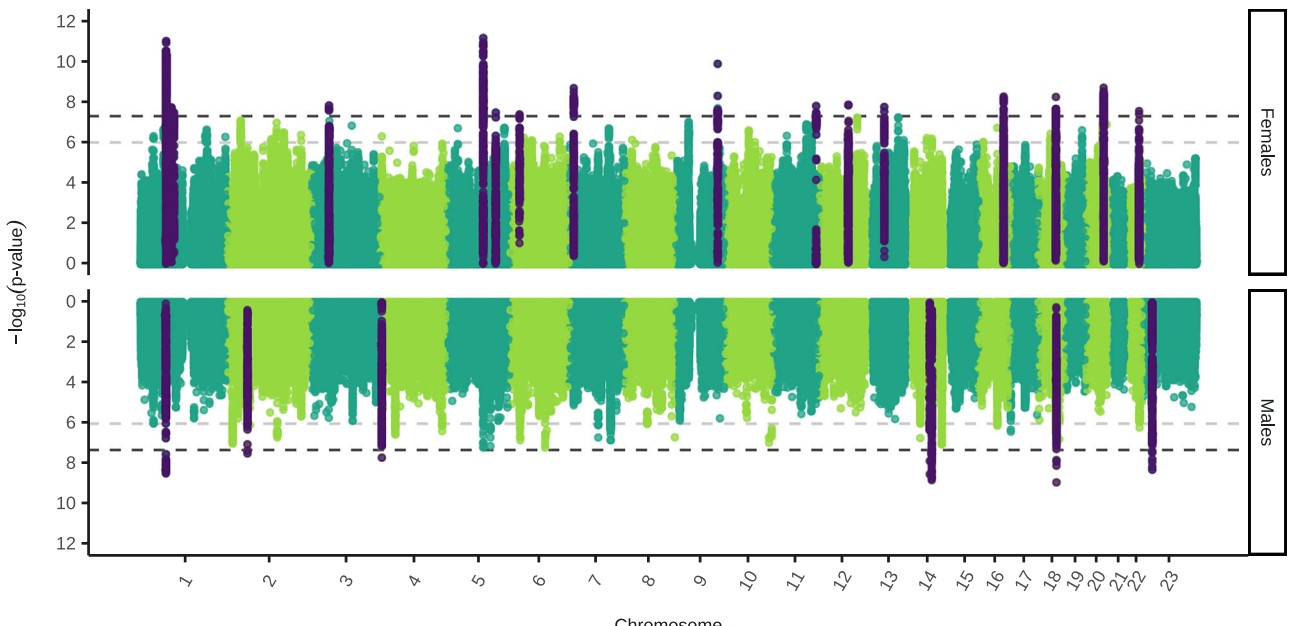

**Fig. 1 | Miami plot of sex-stratified genome-wide association study (GWAS) meta-analysis of Major Depressive Disorder (MDD), with female and male meta-analyses shown on the top and bottom, respectively.** The two-sided, unadjusted −log10 $p$ values for GWAS results of each single nucleotide polymorphism (SNP) are shown with positions according to human genome build 37 (GRCh37 assembly). Chromosome 23 is the X chromosome. The darker grey and lighter grey dotted horizontal lines indicate genome-wide significance ($P = 5 \times 10^{-8}$) and nominal significance ($P = 1 \times 10^{-6}$), respectively. SNPs in dark purple indicate the lead independent genome-wide significant SNPs, and any SNPs in linkage disequilibrium with them. Females: 130,471 cases, 159,521 controls. Males: 64,805 cases, 132,185 controls.

our sensitivity analyses. When the same population prevalence was used for females and males, we found that $h^2_{SNP}$ on the liability scale is similar across sexes, suggesting the sex difference in $h^2_{SNP}$ is driven by the prevalence difference (Supplementary Fig. 3G). Refer to Supplementary Note 1, Supplementary Fig. 3 and Supplementary Data 9,10 for more details.

### Genetic correlation of MDD across sexes

Autosomal SNP-based genetic correlation ($r_g$) estimated from LDSC was used to determine whether there is evidence for sex-specific genetic effects, at the genome-wide level, contributing to MDD risk. There was a significantly higher $r_g$ between the previous female GWAS of MDD by Blokland et al.[16] and our female GWAS than with our male GWAS (Z = 4.49, padj(Benjamini-Hochberg) = $7.3 \times 10^{-6}$). The male GWAS by Blokland et al.[16] showed a significantly higher $r_g$ with our male GWAS than with our female GWAS (Z = 3.23, padj(B-H) = 0.003). A similar, though non-significant, trend was observed when comparing our sex-stratified GWAS to those from Silveira et al.[17] (Supplementary Fig. 1, and Supplementary Data 4–5).

The $r_g$ between females and males was significantly less than one when using our sex-stratified meta-analysis results ($r_g = 0.90 \pm 0.03$, H0: $r_g = 1$ [Z = −3.8, $p_{adj}$(B-H) = 0.0003]) (Fig. 2d). This result suggests that MDD variants are not fully shared across the sexes. To determine whether this $r_g < 1$ reflects sex differences or general heterogeneity across cohorts, we also estimated $r_g$ between each pairwise combination of the two sexes by the six cohorts. Meta-analysis of the $r_g$ estimates from all 30 female-male across cohort comparisons was also significantly less than one ($r_g = 0.70 \pm 0.03$, H0: $r_g = 1$ [Z = −8.6, $p_{adj}$(B-H) = $3.7 \times 10^{-17}$]). We then benchmarked these across-sex $r_g$ estimates against within-sex estimates; meta-analysis of the $r_g$ estimates from all 15 male-male across cohort comparisons was not significantly different from one ($r_g = 0.94 \pm 0.06$, H0: $r_g = 1$ [Z = −1.06, $p_{adj}$(B-H) = 0.36]), but the meta-analysis of all 15 female-female across cohort comparisons was significantly less than one ($r_g = 0.73 \pm 0.04$, H0: $r_g = 1$ [Z = −6.01, $p_{adj}$(B-H) = $4.56 \times 10^{-9}$]). Furthermore, removing cohort variation by conducting a meta-analysis of the $r_g$ estimates from all six female-male within cohort comparisons showed a $r_g$ estimate not significantly different from one ($r_g = 1.02 \pm 0.05$, H0: $r_g = 1$ [Z = 0.30, $p_{adj}$(B-H) = 0.76]). Together, these results suggest that the female-male $r_g$ being significantly less than one could reflect heterogeneity between cohorts (especially among the female datasets) rather than true genome-wide sex differences.

We also investigated whether the MDD effect sizes (betas) are different across the sexes for SNPs known to be associated with MDD. The Pearson correlation was calculated between the standardised beta values of our male and female meta-analysis summary statistics, and between each pairwise combination of the two sexes by six cohorts, for the lead independent genome-wide significant SNPs from the largest GWAS meta-analysis of MDD (sex-combined)[27] (Fig. 2e, and Supplementary Figs. 4–6). The correlation between male and female MDD effect sizes was significantly less than one when using our sex-stratified meta-analysis results (R = $0.81 \pm 0.02$, H0: R = 1 [Z = -9.34, $p_{adj}$(B-H) = $1.23 \times 10^{-20}$]) and for meta-analysis of the 30 male-female across cohort comparisons (R = $0.33 \pm 0.02$, H0: R = 1 [Z = −29.5, $p_{adj}$(B-H) = $4.16 \times 10^{-191}$]). Our benchmarking analyses were also all significantly less than one; meta-analysis of the 15 male-male (R = $0.28 \pm 0.03$, H0: R = 1 [Z = -23.6, $p_{adj}$(B-H) = $9.91 \times 10^{-123}$]) and 15 female-female (R = $0.39 \pm 0.03$, H0: R = 1 [Z = −17.35, $p_{adj}$(B-H) = $3.17 \times 10^{-67}$]) across cohort comparisons. Therefore, the female vs male correlations being significantly less than one could be due to the inherent heterogeneity of MDD rather than sex differences in the MDD effect sizes of SNPs associated with sex-combined MDD risk. However, removing cohort variation by conducting a meta-analysis of the correlations from all six female-male within cohort comparisons showed a correlation significantly less that one (R = $0.39 \pm 0.07$, H0: R = 1 [Z = −8.9, $p_{adj}$(B-H) = $8.06 \times 10^{-19}$]). This suggests that sex differences in genetic effects that are not fully explained by across cohort heterogeneity may exist within cohorts. A similar approach using the slope and intercept from linear regressions of male versus female effect size estimates also found

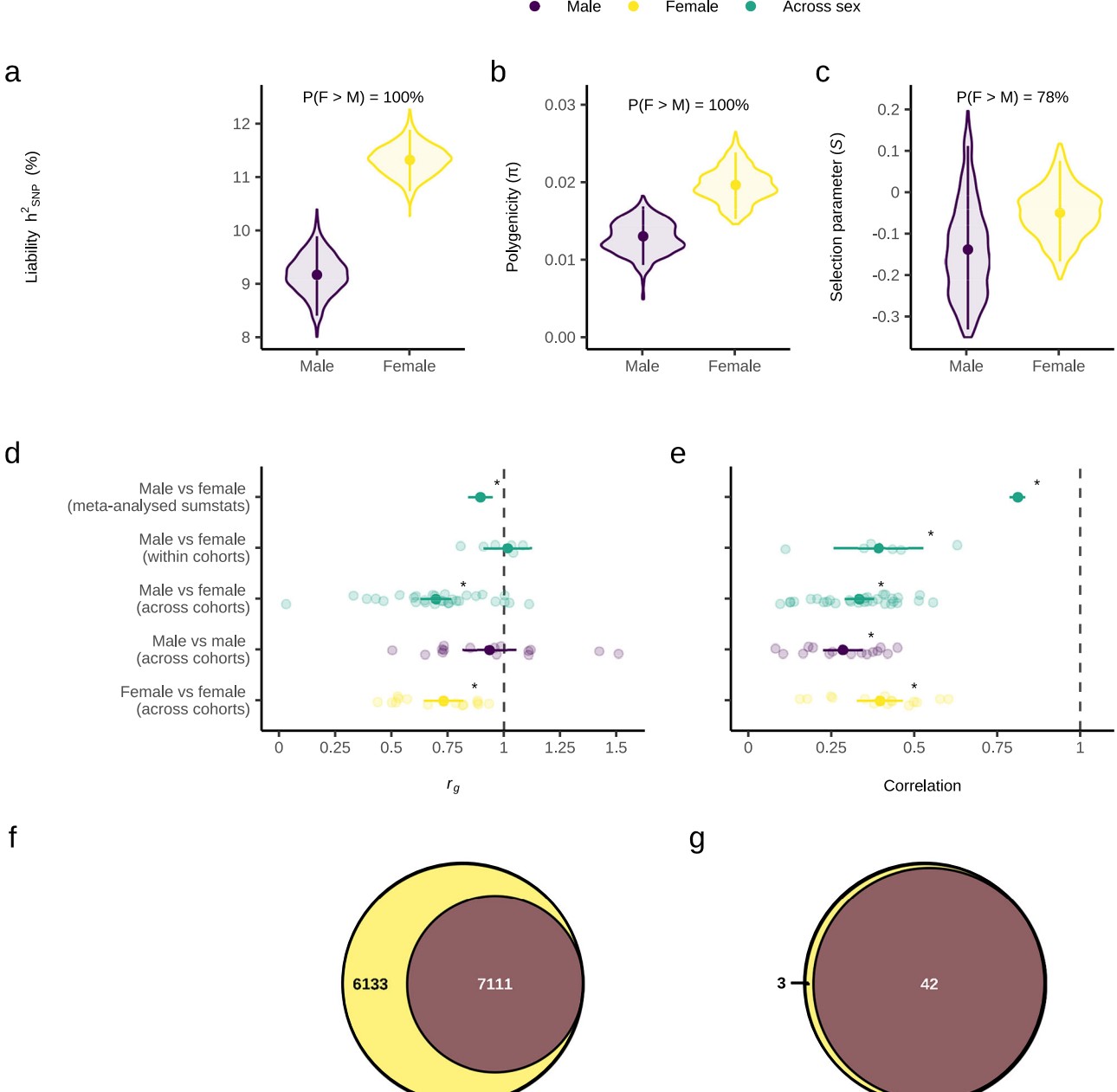

**Fig. 2 | Comparisons of genetic architecture and polygenic overlap across the two sexes. a** Autosomal SNP-based heritability ($h_{SNP}$) on the liability scale using a population prevalence of 0.1 in males and 0.2 in females, **b** Polygenicity, and (**c**) Selection parameter. **d** Autosomal SNP-based genetic correlation ($r_g$) between males and females using our sex-stratified meta-analysis results, and meta-analysis of $r_g$ estimated in all six male-female within cohort, 30 male-female across cohort, 15 male-male across cohort and 15 female-female across cohort combinations. **e** Pearson correlation of the Major Depressive Disorder (MDD) effect sizes of SNPs known to be associated with sex-combined MDD for males vs females using our sex-stratified meta-analysis results, and meta-analysis of Pearson correlations estimated in all six male-female within cohort, 30 male-female across cohort, 15 male-male across cohort and 15 female-female across cohort combinations. **f** Venn diagram depicting the number of causal variants explaining 90% of MDD $h_{SNP}^2$ in females only, males only, or both sexes, as identified by MiXeR. **g** Venn diagram depicting the number of genomic regions that contain a

causal variant for MDD in females only, males only or both sexes, as identified by gwas-pw. Females or female-female comparisons are in yellow, males or male-male comparisons in dark purple and female-female comparisons in green. For **a**–**c** a Bayesian framework was used. Estimates were obtained using SBayesS with summary statistics from the sex-stratified GWAS meta-analysis (females: 130,471 cases and 159,521 controls; males: 64,805 cases and 132,185 controls). Violin plots display the posterior distributions, points represent the mean posterior value, error bars are the 95% highest posterior density interval and percentages are the posterior probability that female value > male value. For **d** and **e** frequentist statistics were used. Background points represent individual values from each study included in the meta-analysis, and overlaid points and error bars represent the mean estimate and 95% confidence interval from the meta-analysis. Stars represent the $r_g$ / R being significantly different to 1, based on a two-sided Z-test with $p$ values adjusted for five comparisons using the Benjamini-Hochberg method. Exact p-values are provided in the Results section.

similar conclusions (Supplementary Note 2, Supplementary Figs. 7-10).

$R_g$ captures the shared genetic architecture across the entire genome whereas the Pearson correlation of effect sizes focuses only on the lead genome-wide significant SNPs previously associated with MDD in a sex-combined GWAS meta-analysis[27]. This distinction may explain why female-male within cohort comparisons showed an $r_g$ estimate not significantly different from one (Fig. 2d), but a correlation significantly less than one (Fig. 2e). The female-male $r_g$ (using our sex-stratified meta-analysis results) being significantly less than one could be driven by between cohort heterogeneity, whereas specific lead SNPs may still exhibit sex-dependent differences in effect sizes.

### Polygenic overlap of MDD across sexes

Bivariate MiXeR[29] revealed that all 7111 (SE = 701) causal variants for MDD in males were shared with MDD in females, with an additional 6,133 (SE = 988) variants unique to MDD in females and zero (SE = 0.0004) variants unique to MDD in males (Fig. 2f). For those causal variants shared by females and males, the correlation of their effect sizes was 1.0 (SE = $6.4 \times 10^{-8}$) (Supplementary Data 11). We also accounted for the differential power across our sex-stratified GWAS by repeating MiXeR analyses using equivalent sample sizes in females and males and found the same pattern (Supplementary Data 12, Supplementary Fig. 3I). We also utilised gwas-pw[30] to identify autosomal regions of the genome containing MDD causal variants that are shared between females and males and regions that are sex-specific. Gwas-pw identified 42 genomic regions that contain a causal variant for MDD shared by both sexes, three regions unique to females and zero regions unique to males, agreeing with the mixer results (Fig. 2g). Within the 42 shared regions, four possible causal risk variants were identified and mapped to multiple genes with two or more methods (*CYSTM1, PFDN1, HBEGF, SLC4A9, TLR4, CTD-2298J14.2, LRFN5, RAB27B*) (Supplementary Data 13). However, no possible causal risk variants were identified within the three female-specific genomic regions (Supplementary Data 14). Both our MiXeR and gwas-pw results suggest that the set of causal variants influencing MDD in males is a subset of those that influence MDD in females.

All 42 shared regions identified by gwas-pw exhibit peaks of MDD association in both sexes (Supplementary Fig. 11). Furthermore, the four possible causal risk variants within these shared regions had concordant effect directions across sexes (Supplementary Fig. 12D). Of the three female-specific regions (gwas-pw), two contain genome-wide significant SNPs in females only and the third includes a SNP nearing significance in females only (Supplementary Fig. 11). This suggests some genome-wide significant SNPs identified in females may be female-specific, not only due to lower male GWAS power. Neither MiXeR nor gwas-pw identified male-specific causal variants/regions. All but one genome-wide significant autosomal SNP in males fell within shared regions (gwas-pw). Thus, SNPs reaching significance in males only may result from stronger effects in males rather than male-specificity, with weaker associations in females going undetected due to power limitations. As gwas-pw and MiXeR only analysed autosomes, male-specific variants may exist on the X chromosome, where one SNP was genome-wide significant in males only.

### Functional annotation and analyses

We mapped genome-wide significant SNPs from our sex-stratified GWAS meta-analysis to genes using positional, eQTL and chromatin interaction mapping (reporting genes supported by two or more methods) (Supplementary Data 15-16), and searched for previous SNP-phenotype associations in GWASCatalog. Female and male genome-wide significant SNPs mapped to different genes, with only one gene, *NEGR1*, mapped in both sexes (Supplementary Data 17). The novel male SNP (rs5971319) has not previously been associated with any

depression phenotypes but has been associated with educational attainment and vaginal microbiome relative abundance (opposite effect direction), and with neuroticism (same effect direction) (Supplementary Data 3 and 18). rs5971319 mapped to the gene *IL1RAPL1* (Supplementary Data 16).

We conducted gene-based, gene-set and gene-property tests for our sex-stratified results in FUMA[31]. 16 genes (female) and 14 genes (male) passed genome-wide significance ($P < 2.53 \times 10^{-6}$), with only *DCC* significant in both sexes (Supplementary Fig. 13). Gene-set analysis using gene-level statistics for all genes revealed that in females, SNPs were significantly enriched for two biological processes; central nervous system neuron development ($p_{Bonferroni} = 0.036$) and central nervous system neuron differentiation ($p_{Bonferroni} = 0.045$). No significant gene sets were found in males. As many gene-set databases emphasise developmental genes, the female enrichments may partly reflect this bias. To explore further, we conducted a gene-property analysis using BrainSpan gene expression data across 11 brain developmental stages. SNPs from both sexes were significantly enriched for expression in the 'late mid-prenatal' stage, with additional 'early mid-prenatal' enrichment in females (Supplementary Fig. 14). These findings support the involvement of neurodevelopmental processes indicated by gene-set analysis. Gene-property analysis for tissue specificity using 30 general tissue types (GTEx v8) identified SNPs from both the female and male stratified GWAS analyses as significantly enriched for gene expression in brain tissue, while pituitary tissue was significantly enriched only in the female SNPs (Supplementary Fig. 15A). Furthermore, using 53 tissues types (GTEx v8), both sexes showed significant enrichment for gene expression in the cortex and frontal cortex, caudate, putamen and nucleus accumbens basal ganglia, hippocampus, amygdala, hypothalamus and anterior cingulate cortex. Only female SNPs were significantly enriched for gene expression in the cerebellum and cerebellar hemisphere (Supplementary Fig. 15B). The differing results likely reflect the power imbalance between our female and male GWAS meta-analyses.

### Genome-wide genotype-by-sex interaction analysis

The sex-stratified GWAS meta-analysis estimated the effect size and direction of SNP associations with MDD in each sex separately. To complement these analyses, we conducted a genome-wide genotype-by-sex interaction (GxS) meta-analysis, which directly tested whether SNP associations with MDD differ significantly between sexes. This GxS meta-analysis combined results from the same five cohorts with those from Blokland et al.[16]. No SNPs reached genome-wide significance, however four independent SNPs were nominally significant ($P < 1 \times 10^{-6}$) (Fig. 3). There was little evidence for residual population stratification (Supplementary Fig. 16). The results for the X chromosome non-pseudoautosomal region were very similar when using full and no dosage compensation (Supplementary Fig. 17). For more details about the GxS analysis, refer to Supplementary Note 3, Supplementary Figs. 12C and 18–22, and Supplementary Data 19–20.

### Sex-specific pleiotropic effects

**Genetic correlations .** We used LDSC to identify genome-wide autosomal SNP-based genetic correlations ($r_g$) between our sex-stratified MDD GWAS meta-analysis results and various psychiatric, metabolic, and substance use traits (Fig. 4a, Supplementary Data 21-23). MDD in females showed significantly higher genetic correlations than MDD in males with sex-combined attention deficit hyperactivity disorder (ADHD) (Z = 3.34, $p_{adj}$(B-H) = 0.003) and with sex-combined regular smoking (Z = 2.85, $p_{adj}$(B-H) = 0.01). There was also a significantly higher genetic correlation between sex-combined metabolic traits (body mass index (BMI) and metabolic syndrome) and MDD in females compared to MDD in males (BMI: Z = 5.64, $p_{adj}$(B-H) = $1.86 \times 10^{-7}$, metabolic syndrome: Z = 4.35, $p_{adj}$(B-H) = $1.38 \times 10^{-5}$). Using sex-stratified BMI GWAS, we found that female MDD had a significantly

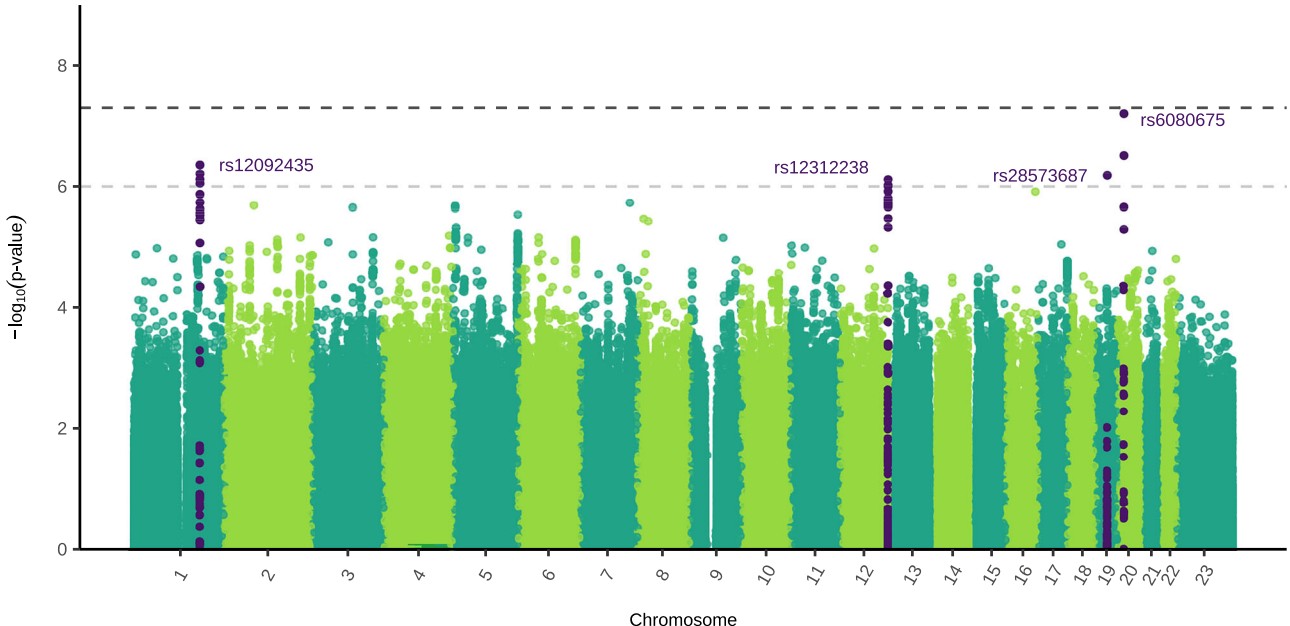

**Fig. 3 | Manhattan plot of genome-wide genotype-by-sex (GxS) interaction meta-analysis for Major Depressive Disorder (MDD).** The two-sided, unadjusted −log10 $p$ values for GxS results of each single nucleotide polymorphism (SNP) are shown with positions according to human genome build 37 (GRCh37 assembly). Chromosome 23 is the X chromosome. The darker grey and lighter grey dotted horizontal lines indicate genome-wide significance ($P = 5 \times 10^{-8}$) and nominal significance ($P = 1 \times 10^{-6}$), respectively. SNPs in dark purple indicate the lead independent nominally significant SNPs, and any SNPs in linkage disequilibrium with them. Females: 130,471 cases, 159,521 controls. Males: 64,805 cases, 132,185 controls.

higher genetic correlation with both female BMI (Z = 5.22, $p_{adj}$(B-H) = 3.56 × 10$^{-7}$) and male BMI (Z = 3.72, $p_{adj}$(B-H) = 0.0002), compared to male MDD (Fig. 4b, and Supplementary Data 24, 25).

**Polygenic overlap .** We used MiXeR to quantify polygenic overlap between MDD in females/males and the metabolic traits BMI and metabolic syndrome (Fig. 4c). The number of causal variants explaining 90% of $h^2_{SNP}$ was estimated to be relatively similar for BMI in females and males (female = 8956 (SE = 232); male = 8415 (SE = 237)), while the number of causal variants estimated for metabolic syndrome (sex combined) was 10,422 (SE = 339) (Supplementary Data 26). Overall, female MDD had larger polygenic overlap with metabolic traits (female MDD-female BMI: $n_{shared}$ = 8549 ± 368; female MDD-sex combined MetS: $n_{shared}$ = 9463 ± 803) than male MDD (male MDD-male BMI: $n_{shared}$ = 4735 ± 689; male MDD-sex combined MetS: $n_{shared}$ = 6078 ± 969) (Supplementary Data 27).

We also utilised gwas-pw to identify sex-specific autosomal genomic regions containing causal risk variants shared between MDD and metabolic traits (Fig. 4d). In both sexes, we identified one genomic region containing a causal variant for both MDD and BMI, which mapped to the genes *FTO* and *IRX3* (Supplementary Data 28). Of the 24 female-specific MDD/BMI pleiotropic regions, possible causal risk loci mapped to multiple genes (*DYNC1I2*, *HTT*, *MSANTD1*, *ANAPC4* and *DENND1A*) (Supplementary Data 29). No male-specific MDD/BMI regions were detected. For MDD and metabolic syndrome, we identified four genomic regions containing a causal variant for both traits shared by both sexes, which also mapped to the *FTO* gene (Supplementary Data 30). Of the 22 female-specific MDD/metabolic syndrome pleiotropic regions, possible causal risk loci mapped to multiple genes (*GPC6*, *SHISA9*, *RP11-154H12.3*, *KCNG2*, *TXNL4A*, *RBFA* and *ADNP2*) (Supplementary Data 31). Lastly, from the four male-specific MDD/metabolic syndrome pleiotropic regions we identified one possible causal risk variant which mapped to the gene *ANKK1* (Supplementary Data 32).

## Discussion

Sex differences in the prevalence and symptomology of Major Depressive Disorder (MDD) are well-documented. Understanding these differences is crucial for uncovering underlying biological mechanisms and developing more targeted treatments[1,4,14,32]. Here, we tested the following hypotheses: genetic variants may contribute to sex differences in MDD via 1) sex-dependent effects in which genetic variants for MDD have differing magnitudes and/or direction of effects across sexes, 2) sex-specific effects in which different genetic variants contribute to MDD across sexes, and/or 3) the presence of genetic variants for MDD on the X chromosome. We also tested the hypothesis that sex-specific pleiotropic effects between MDD and other phenotypes may contribute to sex differences in the phenotypic presentation of MDD. To do this, we conducted the largest sex-stratified genome wide association and genotype-by-sex interaction meta-analyses for MDD to date.

In our GWAS of MDD in males we identified one novel independent genome-wide significant SNP (rs5971319). Notably, this novel SNP is located on the X chromosome and maps to the gene *IL1RAPL1* which is involved in the hippocampal memory system and is linked to intellectual disability[33]. A SNP in linkage disequilibrium (LD) with rs5971319 has previously been associated with educational attainment in the opposite direction, consistent with previous observations of the negative causal relationship between educational attainment and MDD risk[34]. The identification of this novel locus, despite having a substantially smaller sample size than the largest sex-combined MDD GWAS meta-analysis[27] highlights the utility of sex-stratified analyses.

Our results provide evidence for sex differences in the heritability of MDD. Autosomal SNP-based heritability ($h^2_{SNP}$) was higher in females than males, and remained consistent in our sensitivity analyses accounting for the differential power across sexes and for across-cohort heterogeneity. This result reflects previous findings in a sex-stratified GWAS[16] and some twin studies[22–24], and suggests there may be a greater genetic contribution to MDD risk in females or

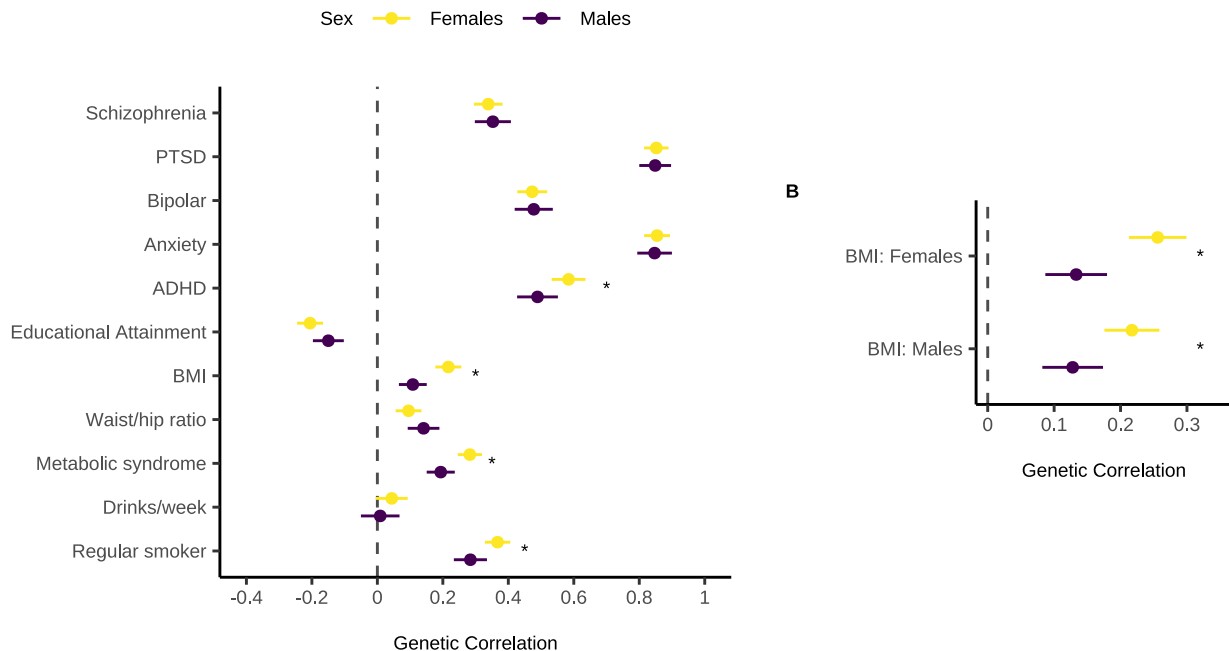

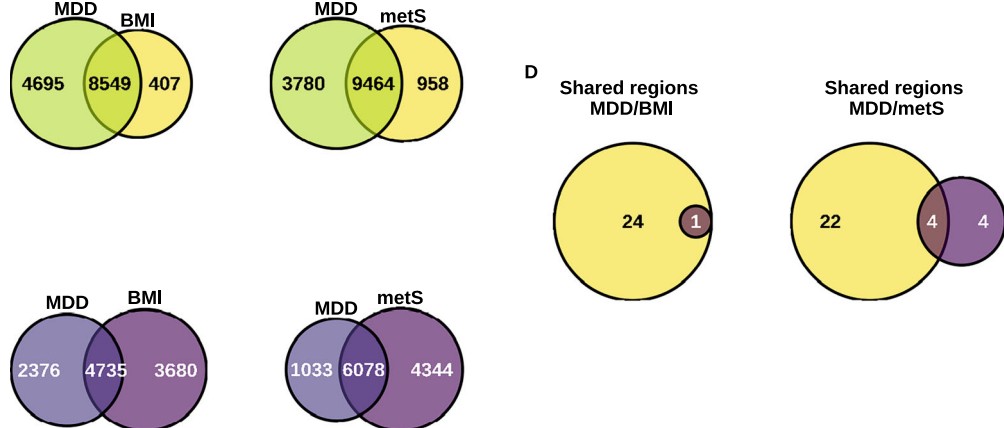

**Fig. 4 | Genetic correlations ($r_g$) and polygenic overlap to identify sex-specific pleiotropy. a** $R_g$ of Major Depressive Disorder (MDD) in females and males with other psychiatric disorders, metabolic and substance use traits. **b** $R_g$ of MDD in females and males with BMI in females and males. **c** Venn diagrams depicting the number of causal variants explaining 90% of $h^2_{SNP}$ in MDD only, BMI/metS only, or both traits as identified by MiXeR in females (yellows) and males (purples). **d** Venn diagrams depicting the number of genomic regions that contain a causal variant for both MDD and BMI/metS in females only, both sexes, or males only as identified in gwas-pw. Females are in yellow and males in dark purple. $R_g$ in panels **a**, **b** were estimated using linkage disequilibrium score regression (LDSC) using our sex-stratified GWAS summary statistics (Females: 130,471 cases, 159,521 controls. Males:

64,805 cases, 132,185 controls) and publicly available GWAS datasets for the other traits (Supplementary Data 21). Points represent the $r_g$ point estimates and error bars denote the 95% confidence interval. Stars indicate a significant difference in $r_g$ between females and males for a given trait, assessed using the jack-knife method and a two-sided Z-test on the difference in $r_g$ across the 200 jack-knife pseudo-values. $P$ values were adjusted using the Benjamini Hochberg method for 11 tests (all traits, panel **a**) or two tests (sex-specific BMI, **b**). Exact $p$ values are provided in Supplementary Data 23 and 25. PTSD = post-traumatic stress disorder, ADHD = attention deficit hyperactivity disorder, BMI = body mass index, Waist/hip ratio = waist to hip ratio adjusted for BMI, metS = metabolic syndrome.

alternatively a stronger environmental influence on MDD risk in males. We found that the sex difference in $h^2_{SNP}$ may be driven by the prevalence difference, as when the same population prevalence was used across sexes $h^2_{SNP}$ on the liability scale was similar in females and males. It is possible that MDD must be more severe in males compared with females for an individual to cross the liability threshold and for MDD to manifest. This increased male severity could explain the lower

prevalence of MDD in males. As more severe phenotypes tend to have a higher heritability[35], it could also explain the disappearance of the sex difference when $h^2_{SNP}$ was calculated on the liability scale assuming equal population prevalence across sexes. Another explanation for the lower prevalence in males could be due to under-reporting and under-diagnosis of MDD in males[7,36]. If this is the case, the true male population prevalence may be higher than reported and controls may

include unidentified cases, resulting in an underestimation of male $h^2_{SNP}$[37]. However, in our analyses $h^2_{SNP}$ remained higher in females even when accounting for up to 30% unscreened male controls, and a corresponding increase in male population prevalence, when compared to females with a population prevalence of 20% and no unscreened controls. Thus, $h^2_{SNP}$ could be higher in females even with underreporting and under-diagnosis in males, although further research is needed to quantify this under-diagnosis.

The origin of the MDD prevalence difference across sexes could also affect interpretation of our heritability findings. If the higher prevalence of MDD in females is predominantly driven by environmental factors, such as trauma and structural forms of discrimination[8,9], the sex difference in $h^2_{SNP}$ may arise from the statistical relationship between prevalence and liability scale $h^2_{SNP}$ rather than underlying sex differences in genetic architecture. However, if the prevalence difference is a consequence of genetic or biological factors, the difference in $h^2_{SNP}$ could indeed signal genuine sex differences in genetic architecture. Importantly, the presence of gene–environment correlations mean that differences in prevalence driven by environmental exposures may still partially reflect underlying genetic effects. Beyond heritability, we found evidence for a higher polygenicity for MDD in females compared to males. Our polygenic overlap analyses also showed that MDD causal variants in males are a subset of causal variants in females. These results remained consistent in our sensitivity analyses accounting for the differential power across sexes. This suggests that the higher $h^2_{SNP}$ in females may reflect, at least in part, distinct genetic architectures across sexes. Overall, our results suggest sex differences in the genetic architecture of MDD with a greater genetic burden for MDD risk in females, which may be driven by a higher number of MDD causal variants and the presence of female-specific SNPs, supporting the sex-specific hypothesis.

We found substantial overlap in genetic variants associated with MDD between male and female MDD. Genetic variants that influence MDD in both sexes had a near-perfect positive correlation. On the other hand, we found both the genetic correlation of our sex-stratified meta-analysis and the correlation of the male and female effect sizes of SNPs known to be associated with MDD to be significantly below one. However, benchmarking all our male-female comparisons with male-male and female-female comparisons demonstrated that these results could be due to the inherent heterogeneity of MDD rather than sex differences. A genetic correlation not significantly different to one was found by Blokland et al.[16] but not by Silveira et al.[17] The discrepancy between our genetic correlation results (not indicating sex differences) and polygenic overlap results (providing evidence for a set of female-specific SNPs) can be explained by the differing methods. Genetic correlation captures the average effect and direction of pleiotropy across the entire genome, but does not capture variation in correlation at specific loci. Thus, the genetic correlation estimate can be unaffected by a set of trait-specific variants if the majority of variants are near identical in effect size and direction (as is the case here where shared male-female variants were correlated at one)[29]. Overall, these results suggest that the greater genetic contribution to MDD risk in females may not be due to effect size differences but rather due to a set of female-specific SNPs, i.e., our results do not support the sex-dependent hypothesis but do support the hypothesis that sex-specific genetic effects contribute towards sex differences in MDD. Polygenic risk score (PRS) analyses may provide another avenue for future research to assess whether the genetic architecture of MDD differs between sexes. However, it will be important to account for the differential power of sex-stratified GWAS used to construct PRS[38].

Despite these similarities, we found very little overlap in genes associated with MDD between sexes. Both our gene-based tests and the annotation of genome-wide significant SNPs identified only one gene significantly associated with MDD in females and males; *DCC* (gene-based test) and *NEGR1* (SNP annotation). Both genes are involved in neuronal connectivity and genetic variants in these and related genes have previously been associated with psychiatric disorders including MDD[39–41]. Due to the differential power of our sex-stratified GWAS meta-analysis, it is difficult to determine whether any of the observed gene associations reflect sex-dependent or sex-specific effects. Our polygenic overlap results suggest the presence of shared and female-specific causal variants, but no male-specific causal variants. The genes identified only in females may be female-specific, but this should be interpreted with caution, as the smaller male GWAS sample may have lacked the power to detect these SNPs identified in females. Conversely, the genes identified only in males may not be male-specific, but rather sex-dependent with stronger effects in males. This could explain why they were detected despite the lower power of the male GWAS.

Females show a higher prevalence of atypical and immuno-metabolic depression, characterised by symptoms such as weight gain, increased appetite and hypersomnia as well as immune-inflammatory pathophysiology, metabolic dysregulation and an increased cardiometabolic disease risk[42–44]. We observed stronger genetic correlations and greater polygenic overlap between MDD in females and metabolic traits (body mass index and metabolic syndrome) than MDD in males with these same traits. Similarly, Silveira et al.[17] found genetic correlations between depression and metabolic features (including body mass index, waist-to-hip ratio and triglycerides), as well as coronary artery disease, were larger and significant only in females. Many of the genes we identified as shared by MDD and metabolic traits in females only are associated with neurological disorders such as epilepsy and Huntington's disease, as well as autism. Our results suggest that sex-specific pleiotropy, and thus sex-specific shared pathophysiological mechanisms, may contribute to the sex differences in metabolic symptoms and comorbidities of people with MDD.

Comorbid substance use and MDD show a range of sex differences. Across studies, alcohol use is consistently higher in males than females with MDD[45]. However, the findings on smoking are mixed; some studies find smoking is higher in males with MDD and others demonstrate it is higher in females[46]. We found a significantly higher genetic correlation between regular smoking and MDD in females than in males, consistent with Silveira et al.[17] This suggests that shared pathophysiological mechanisms may contribute to the comorbidity of smoking and MDD to a larger extent in females. We found a non-significant genetic correlation between MDD in both females and males with alcohol use (drinks per week). This suggests that the comorbidity of MDD and alcohol use, and its sex difference, may largely be driven by environmental factors rather than shared pathophysiological mechanisms.

The heterogeneity of MDD across sexes may also be influenced by the environment and gene-by-environment interactions. Genetic predisposition, trauma, and their interaction have been shown to influence MDD risk[47–50]. Furthermore, genome-by-trauma interaction effects on MDD were larger in males than females[51]. Interestingly, one of the nominally significant SNPs identified in our genotype-by-sex interaction meta-analysis has previously been associated with MDD with trauma exposure (rs11671136 in LD with rs28573687)[47]. Future work examining whether sex-specific gene-by-environment interactions also contribute to sex differences in MDD will likely be informative. Our sex-stratified summary statistics published here will provide a valuable resource for future analyses.

Our study should be interpreted considering some limitations. Firstly, the genome-wide meta-analysis of MDD has a 1.65-fold larger effective sample size in females compared to males ($n_{eff\ (Females)} = 287,082$; $n_{eff\ (Males)} = 173,943$) and this power difference could exaggerate sex differences identified. Secondly, our analyses are

restricted to Europeans only limiting the applicability of our findings to other populations. Thirdly, as the software used to conduct our genotype-by-sex interaction analysis only includes the genotype-by-sex interaction term, and cannot include all genotype-by-covariate and sex-by-covariate interaction terms, confounders may not have been properly controlled for in this analysis[52]. Lastly, quality control of the genotype data was performed prior to this study and, as such, most recommended sex-aware quality control practices[38] were not implemented, potentially introducing undetected sex-specific technical biases.

Here, we conducted the largest sex-stratified genome wide association and genotype-by-sex interaction meta-analyses for MDD to date. Our findings reveal a novel genetic variant associated with MDD in males, and provide evidence for a higher burden of genetic risk in females. We highlight the potential role of the X chromosome in modulating MDD risk differently between sexes. These insights enhance our understanding of the genetic basis for sex differences in MDD and underscore the importance of sex-stratified approaches in genetic research. Moving forward, integrating sex-specific genetic findings into clinical practice could pave the way for more personalised diagnostic and therapeutic strategies for MDD. For example, these results may inform the development of novel therapeutics that target sex-dependent biological pathways, ultimately improving treatment efficacy and outcomes.

## Methods

### Ethics statement
This study complies with all relevant ethical regulations for research involving human participants and was conducted in accordance with the principles of the Declaration of Helsinki. The meta-analysis was performed using previously collected data from individual cohorts. Ethics approval and informed consent were obtained for all original studies by their respective institutional review boards or ethics committees, as detailed in Supplementary Methods 1.

### Participants
Data from five international cohorts were analysed: The Australian Genetics of Depression Study (AGDS) in Australia[53] (Female: cases = 10,406, controls = 7147; Male: cases = 3174, controls = 6601), The BIObanks Netherlands Internet Collaboration (BIONIC) in the Netherlands[54] (Female: cases = 10,664, controls = 26,878; Male: cases = 4432, controls = 20,013), the Genetic Links to Anxiety and Depression (GLAD + ) study in the United Kingdom (UK)[55] (Female: cases = 16,708, controls = 3393; Male: cases = 4656, controls = 3025), the UK Biobank from the UK[56] (Female: cases = 46,194, controls = 53,211; Male: cases = 22,608, controls = 56,516) and All Of Us from the United States of America[57] (Female: cases = 26,776, controls = 46,678; Male: cases = 11,136, controls = 35,836). Across all cohorts, MDD cases and controls were defined primarily based on the DSM (Diagnostic and Statistical Manual of Mental Disorders) where available, supplemented by electronic health records and/or self-report of diagnosis (Supplementary Data 1). Sex was defined by chromosomal composition, with XX individuals classified as female and XY as male. We have complied with all relevant ethical regulations and informed consent was obtained from all participants. See Supplementary Methods 1 for more details on each cohort and Supplementary Data 33 summarises how the recommended best practices for sex-aware analyses from Khramtsova et al.[38] were addressed.

### Genotyping, quality control and imputation
Details about genotyping platform used as well as quality control and imputation methods can be found in Supplementary Methods 1. In all cohorts, individual quality control included exclusion of participants with non-European ancestry.

### Association Analyses
All association analyses in all cohorts were conducted using fastGWA in GCTA v1.94.1[58]. Sex-stratified genome-wide association studies (GWAS) were carried out in each cohort on the autosomes and X chromosome using a mixed linear model for a binary outcome (--fastGWA-mlm-binary) with a sparse genetic relationship matrix and the first 10 ancestry principal components, as well as any cohort-specific covariates. Genome-wide genotype-by-sex interaction (GxS) analyses were also carried out in each cohort using a mixed linear model (--fastGWA-mlm) with a sparse genetic relationship matrix, sex as the environment variable and the first 10 ancestry principal components and any cohort-specific covariates. GxS analyses were conducted for the autosomes, as well as for the X chromosome with the non-pseudoautosomal region analysed both with no dosage compensation (--dc 0) and full dosage compensation (--dc 1). The sex-stratified GWAS and GxS results were filtered to retain SNPs with a minor allele frequency larger than 0.01 and a $R^2$ imputation score larger than 0.6. See Supplementary Methods 1 for cohort specific details. The sex-stratified GWAS was used to characterise the SNP effect sizes and directions of effect in each sex separately. These summary statistics were also leveraged for downstream analyses, such as SNP-based heritability, genetic correlation and polygenic overlap across sexes. The GxS analysis complements the sex-stratified GWAS by formally testing whether the differences in SNP effect sizes between sexes are statistically different.

### Meta-analysis
The sex-stratified GWAS and GxS results from the five cohorts were meta-analysed with summary statistics from Blokland et al.[16] which resulted in a total sample size of 130,471 cases and 159,521 controls in females, and 64,805 cases and 132,185 controls in males. As the GxS association analyses in all studies used linear mixed models the beta values were converted to the logistic scale by firstly converting the beta values to odds ratios (OR) using the R function from Lloyd-Jones et al.[59] and then using the log of the OR. The standard error of the log(OR) was calculated as the log(OR) divided by the Z-score. Standard error weighted meta-analyses without genomic control were run in METAL (released 05/05/2020)[60]. The meta-analysis results were filtered to only retain SNPs present in three or more cohorts (at least 50% of the cohorts). In order to identify lead, independent SNPs, clumping was carried out in PLINK v1.90b6.8[61,62] with a linkage disequilibrium threshold (--clump-r2) of 0.1 and a physical distance threshold (--clump-kb) of 1000.

### Replication cohort
Generation Scotland served as a replication cohort[63]. Case status was determined by case classification in the Structured Clinical Interview for DSM-IV disorders (SCID)[64,65] or the Composite International Diagnostic Interview (CIDI)[66]. Individuals also present within the UK Biobank and of non-European ancestry were removed, resulting in a sample size of 2441 cases and 3321 controls in females, and 938 cases and 2491 controls in males. Sex-stratified association analyses were run using fastGWA in GCTA v1.94.1[58] (--fastGWA-mlm-binary) with a sparse genetic relationship matrix and the first four ancestry principal components as covariates. We compared the beta-values from the lead independent genome-wide significant SNPs from our meta-analysis to the replication analysis. For females all 16 lead SNPs were tested and seven lead SNPs were tested in males (one SNP was unavailable in the Generation Scotland cohort). A one-sided binomial test was used to determine whether significantly more than 50% of the SNPs tested in the replication analysis (expected by chance) had a concordant effect size direction. See Supplementary Methods 1 for more details on the Generation Scotland cohort.

## GxS sensitivity analyses

As interaction analyses require very high power in order to detect significant interactions, we also carried out a sensitivity analysis to determine whether any SNPs known to be associated with sex-combined MDD had a significant GxS interaction. We used the genome-wide significant SNPs from the largest MDD GWAS meta-analysis by Adams et al.[27] and a P-value threshold of $7.17 \times 10^{-5}$ (0.05/697 independent SNPs).

## Genetic architecture of MDD in females and males

Autosomal SNP-based heritability ($h^2_{SNP}$), polygenicity (π) and the selection parameter ($S$), a measure of the relationship between effect size and minor allele frequency which indicates the direction and strength of natural selection, were estimated using our sex-stratified meta-analysis summary statistics in SBayesS using GCTB v2.5.2[67]. The linkage disequilibrium shrunk sparse matrix of 2.8 million variants[68] and four chains of length 25,000 with a burn-in of 5000 and thinning of 10 were used. $H^2_{SNP}$ was converted to the liability scale using the method from Lee et al.[69] with a population prevalence of 0.2 and 0.1 in females and males, respectively. To determine whether $h^2_{SNP}$, π and $S$ are different across sexes, we calculated the posterior probability that female value > male value by counting the frequency of Markov chain Monte Carlo (MCMC) samples in which female value > male value. We conducted a range of sensitivity analyses comparing the genetic architecture of MDD in females and males (Supplementary Methods 2). This included calculating $h^2_{SNP}$ estimates in LDSC, accounting for the differential power of our sex-stratified GWAS meta-analysis, as well as examining the role of male under-diagnosis and across-cohort heterogeneity.

## Genetic correlation

Linkage Disequilibrium Score Regression (LDSC) (released 13/02/2015)[70,71] with European LD scores computed from 1000 Genomes was used to estimate bivariate autosomal SNP-based genetic correlations ($r_g$). We estimated $r_g$ between our sex-stratified summary statistics with the largest GWAS meta-analysis of sex-combined MDD (including 23andMe, Inc and Europeans only)[27] and previous sex-stratified GWAS of MDD[16,17]. We also estimated $r_g$ between our male and female meta-analysis summary statistics. To account for across-cohort heterogeneity we also estimated $r_g$ between each pairwise combination of the two sexes by six cohorts (the five new cohorts and the existing GWAS meta-analysis from Blokland et al.[16] used in our meta-analyses here).

We also investigated whether the MDD effect sizes (betas) for SNPs known to be associated with sex-combined MDD are different across the sexes. We entered all of the genome-wide significant hits associated with MDD from Adams et al.[27] (Europeans only) into PLINK v1.90b6.8[61,62] clumping with a linkage disequilibrium threshold (--clump-r2) of 0.1 and a physical distance threshold (--clump-kb) of 1000 to identify lead, independent SNPs. For these SNPs, ensuring the minor allele was the tested allele and consistent across summary statistics, the Pearson correlation was calculated between the standardised effect size estimates (beta values) of our male and female meta-analysis summary statistics, and between each pairwise combination of the two sexes by six studies. Beta values and their standard errors were standardised accounting for the influence of allele frequency and sample size (Eqs. (1) and (2)), where Z represents the Z statistic, $p$ the allele frequency and $n$ the sample size.

$$\beta_{std} = \frac{Z}{\sqrt{2p(1-p)(n+Z^2)}} \tag{1}$$

$$SE_{std} = \frac{1}{\sqrt{2p(1-p)(n+Z^2)}} \tag{2}$$

For both the $r_g$ estimates and Pearson's correlations, an inverse-variance weighted meta-analysis using a random-effects model (restricted maximum likelihood estimator) in R v4.3.1[72] and the 'metafor' package[73] was completed for the six male-female comparisons within studies, the 30 male-female comparisons across studies, the 15 male-male comparisons across studies and the 15 female-female comparisons across studies (same sex within study comparisons were not included as they would yield a correlation of 1) (Supplementary Figs. 4–6). Within-sex comparisons were included as a baseline check to determine whether male-female differences were due to across-sex or across-study heterogeneity. To determine whether $r_g$ estimates / Pearson correlations were significantly different from one we calculated the Z-score, where $H_o$ = the null hypothesis (1 in this case) (Eq. (3)). The P-value was calculated from a standard normal distribution using a two-tailed test and adjusted for five comparisons using the Benjamini-Hochberg method.

$$Z = \frac{(r_g - H_0)}{SE} \tag{3}$$

A similar approach was repeated using the slope and intercept from linear regressions of male versus female effect size estimates (see Supplementary Methods 3).

## Polygenic overlap of MDD across sexes

We used univariate and bivariate MiXeR v1.3[28,29] to quantify polygenicity of male and female MDD and estimate polygenic overlap between sexes. MiXeR fits a Gaussian mixture model assuming that common genetic effects on a trait are a mixture of causal variants and non-causal variants. Polygenicity is reported as the number of causal variants that explain 90% of $h^2_{SNP}$ (to avoid extrapolating model parameters into the area of infinitesimally small effects). As a sensitivity analysis to account for the differential power of our sex-stratified GWAS, we also ran univariate and bivariate MiXeR on the full UK Biobank sample ($n_{Females}$ = 46,194 cases and 53,211 controls, $n_{Males}$ = 22,608 cases and 56,516 controls) and after down-sampling ($n$ = 22,608 cases and 53,211 in both females and males).

We used gwas-pw v0.21[30] to identify causal risk loci for MDD that are sex-specific or shared across the sexes. The recommended European bed file that splits the genome (autosomes only) into approximately independent LD blocks was used within gwas-pw and correlation was set as zero because our female and male MDD GWAS do not have any overlapping individuals. Genomic regions with a posterior probability of association (PPA) larger than 0.5 for model one, two and three were identified as regions containing a causal variant for MDD in females only, males only or shared by both sexes, respectively. Subsequently, for each of these identified regions the SNP with the largest PPA that was above 0.5 was selected as a possible causal MDD variant. These possible causal variants were then annotated using SNP2GENE from FUMA v1.5.2[31] with ANNOVAR (2017-07-17) and ensemble v110. Positional mapping was carried out with a window size of 10 kb, eQTL mapping using all the tissue types from TIGER[74], InSPIRE[75], EyeGEx[76], eQTL catalogue, PsychENCODE[77], van der Wijst et al.[78] scRNA eQTLs, DICE[79], eQTLGen, Blood eQTLs[80], MuTHER[81], xQTLServer[82], CommonMind Consortium[83], BRAINEAC[84] and GTEx v8[85], and 3D Chromatin Interaction Mapping using all Buildin chromatin interaction data (Hi-C of 21 tissue/cell types from GSE87112[86], Hi-C loops from Giusti-Rodríguez et al.[87], Hi-C based data from PsychENCODE[77] and Enhancer-Promoter correlations from FANTOM5[88–90]). All of the annotation datasets from PsychENCODE, FANTOM5 and Brain Open Chromatin Atlas[91] were used for all three mapping types. Genes annotated to a SNP with more than one method were considered.

## Functional annotation and analyses

Gene-based, gene-set and gene-property tests were carried out for the sex-stratified and GxS results using MAGMA v1.08[92] within the SNP2GENE function in FUMA v1.5.2[31]. Our summary statistics as well as the lead, independent SNPs as determined by PLINK clumping were used as inputs. Settings used were ensembl v110, ANNOVAR (2017-07-17)[93], maximum $P$ value of lead SNPs and $P$ value cut-off of $5 \times 10^{-8}$ (sex-stratified summary statistics) or $1 \times 10^{-6}$ (GxS summary statistics), a first and second $R^2$ threshold of 0.1, the European 1000 Genomes Phase 3 reference panel[94], and a gene window size of 10 kb. The gene-based test used a genome-wide significance $P$ value of $2.53 \times 10^{-6}$ as the input SNPs mapped to 19,759 protein coding genes. The gene-set analysis used the gene-level association statistics for all genes from the MAGMA gene-based test and 10,678 gene sets (4761 curated gene sets and 5917 Gene Ontology terms) from MsigDB v6.2[95] with significance determined using Bonferroni corrected $P$ values. A competitive gene-set analysis was used which tests whether genes in a given gene set are more strongly associated with MDD than the remaining genes not in this gene set, and potential confounding factors such as gene size, gene density and linkage disequilibrium are accounted for. Gene property analysis for tissue specificity was conducted to determine whether genes that are more highly expressed in specific tissues tend to be more associated with MDD. We used RNA-sequencing data from 30 general tissue types and 53 tissue types from GTEx v8[85], as well as from brain samples collected at 11 general developmental stages from BrainSpan[96]. Genome-wide significant SNPs (sex-stratified analysis) and nominally significant SNPs (GxS analysis) underwent gene annotation using positional, eQTL and 3D Chromatin Interaction mapping using the same settings as specified above (*'Polygenic overlap of MDD across sexes'*).

To determine whether any of our genome-wide significant SNPs (sex-stratified analysis) and nominally significant SNPs (GxS analysis) are novel and to search for previous SNP-phenotype associations, a second analysis using the SNP2GENE function in FUMA v1.5.2 with GWASCatalog (e0_r2022-11-29) was run. The settings as described above were used, however the maximum $P$ value cut-off was changed to one, and SNPs from the reference panel were included. A lead independent significant SNP was considered novel if the significant SNP, or any SNPs in linkage disequilibrium with it, had not previously been associated with any depression phenotypes according to the list of SNPs in GWASCatalog. As the results from the largest GWAS meta-analysis of MDD[27] had not yet been added to GWASCatalog at the time of our analyses, all significant SNPs, and those in LD with them, were also checked against all of the genome-wide significant SNPs, and any SNPs in LD, in this publication. SNPs in LD were determined using PLINK v1.90b6.8[61,62] with a linkage disequilibrium threshold (--clump-r2) of 0.1 and a physical distance threshold (--clump-kb) of 1000.

## Sex-specific pleiotropic effects

We used LDSC (released 13/02/2015) to estimate genome-wide autosomal SNP-based $r_g$ between our sex-stratified MDD GWAS meta-analysis results and a range of other psychiatric disorders, metabolic traits and substance use traits (Supplementary Data 21). Metabolic and substance use traits were included to determine whether sex-specific pleiotropic effects may contribute to sex differences in metabolic symptoms and substance use in people with MDD. As body mass index (BMI) has a well-powered sex-stratified GWAS[97], we also estimated $r_g$ for all bivariate combinations of our sex-stratified MDD GWAS and the sex-stratified BMI summary statistics.

To determine whether $r_g$ between MDD and each trait is significantly different across sexes we used the jack-knife method, a resampling technique which systematically leaves out one block of data at a time. Unlike the Z-score method, which is commonly used, the jack-knife method accounts for LD structure and does not assume independence of the two genetic correlation values, which is

important as the same second trait is used in both the genetic correlations being compared (e.g., MDD in females versus sex-combined ADHD compared to MDD in males versus sex-combined ADHD). In LDSC, the --print-delete-vals flag was used to obtain delete values of genetic covariance and heritabilities for 200 blocks, with delete values referring to estimates computed by leaving out one of 200 jack-knife blocks. For each bivariate genetic correlation, delete values of genetic correlation were calculated for each of the 200 blocks (Eq. (4)), where $G$ = genetic covariance and $h^2$ = SNP-based heritability.

$$r_g = \frac{G}{\sqrt{h^2(\text{trait1}) \cdot h^2(\text{trait2})}} \tag{4}$$

Jack-knife pseudo-values were then calculated for each of the 200 blocks (Eq. (5)), where $P$ = jack-knife pseudo-value, $n$ = total number of jack-knife blocks (200 in this case), $r_{g,\,\text{global}}^{(1)}$ = the global genetic correlation obtained from LDSC for comparison one, $r_{g,\,\text{global}}^{(2)}$ = the global genetic correlation obtained from LDSC for comparison two, $r_{g,\,\backslash i}^{(1)}$ = the delete genetic correlation estimated with block $i$ removed from comparison one, and $r_{g,\,\backslash i}^{(2)}$ = the delete genetic correlation estimated with block $i$ removed from comparison two.

$$P = n \cdot \left( r_{g,\,\text{global}}^{(1)} - r_{g,\,\text{global}}^{(2)} \right) - (n-1) \cdot (r_{g,\,\backslash i}^{(1)} - r_{g,\,\backslash i}^{(2)}) \tag{5}$$

The mean and standard error of the 200 jack-knife pseudo-values estimates the difference between the two genetic correlations and was compared to the null hypothesis that there is no difference between the genetic correlations by calculating the Z-score, where $H_o$ = the null hypothesis (0 in this case) (Eq. (3)). The P-value was calculated from a standard normal distribution using a two-tailed test followed by the Benjamini Hochberg correction for 11 tests (all traits) or two tests (sex-specific BMI). As a sensitivity analysis, the Z-score method was also used to determine whether $r_g$ between MDD and each trait is significantly different across sexes because although it is not theoretically appropriate it is commonly used (Supplementary Methods 4).

We further explored the metabolic traits BMI and metabolic syndrome that showed significant sex-specific $r_g$. To quantify polygenic overlap between MDD in females/males and each metabolic trait, we used MiXeR v1.3[28,29]. Gwas-pw v0.21[30] was used to identify causal risk loci shared by MDD and each metabolic trait that are sex-specific or shared across the sexes. Inputs to gwas-pw were our female MDD summary statistics and female BMI summary statistics[97], or our male MDD summary statistics and male BMI summary statistics[97], and the recommended European PLINK bed file that splits the genome (autosomes only) into approximately independent LD blocks. The correlation between beta values of SNPs within genomic regions with a PPA < 0.2 as calculated in fgwas v0.3.6[98] was used to account for potential overlapping of cohorts. Genomic regions with a PPA > 0.5 for model 3 were compared between the female MDD–female BMI and male MDD–male BMI analyses to identify genomic regions with a causal variant shared by MDD and BMI that are female-specific, male-specific or shared by both sexes. Subsequently, for each of these identified regions the SNP with the largest PPA that was above 0.5 was selected as a causal MDD/BMI variant. These causal variants were then annotated using SNP2GENE from FUMA v1.5.2[31] with the same settings as described above (*'Polygenic overlap of MDD across sexes'*). The above was repeated for the comparison of the sex-combined metabolic syndrome summary statistics[99] with our female and male MDD summary statistics.

## Reporting summary

Further information on research design is available in the Nature Portfolio Reporting Summary linked to this article.

## Data availability

The raw genotype and phenotype data used to run the association analyses in all separate cohorts are protected and are not available due to data privacy laws. The GWAS meta-analysis summary statistics generated in this study have been deposited in the GWAS Catalogue database under accession numbers GCST90565869, GCST90565870, GCST90565871, and GCST90565872. The source data for figures in the main text are available from Github (https://github.com/jodithea/Sex_differences_genetics_depression), which has been archived on Zenodo and assigned a DOI: 10.5281/zenodo.15233098 (https://doi.org/10.5281/zenodo.15233098)[100].

## Code availability

Code used to conduct analyses presented in this manuscript can be found at Github (https://github.com/jodithea/Sex_differences_genetics_depression), which has been archived on Zenodo and assigned a DOI: 10.5281/zenodo.15233098 (https://doi.org/10.5281/zenodo.15233098)[100].

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

## Acknowledgements

We would like to thank the research participants from each of the cohorts for giving their time to make this work possible. We also thank the employees from All of US and the UK Biobank. *The Australian Genetics of Depression Study (AGDS):* AGDS was primarily funded by the National Health and Medical Research Council (NHMRC) of Australia Grant No. 1086683, and the QSkin study was funded by the NHMRC (Grant Numbers 1185416, 1063061 and 1073898). We thank everyone who contributed to the conception, implementation, media campaign and data cleaning of the AGDS project, including Richard Parker, Simone Cross, and Lenore Sullivan. Thank you to Scott Gordon for carrying out the imputation and quality control of the AGDS and QSkin data and Penelope Lind for contributing to the management of the AGDS database. This work was supported by the following NHMRC Investigator Grants; 2027002 (J.G.T.), 2017176 (B.L.M.); 1172917 (S.E.M.), 1172990 (N.G.M.), 1173790 (N.R.W.), 2026567 (D.C.W.) and 2016346 (I.B.H.). It was also supported by funding from the NHMRC-funded PRE-EMPT Centre of Research Excellence (E.M.B.). *UK Biobank:* The UK Biobank Resource (application number 25331) was used for this work. *BIONIC:* To create the BIONIC consortium we are grateful to for funding from the Biobanking and Biomolecular Resources Research Infrastructure (BBMRI-NL: 184.021.007; 184.033.111) (D.I.B. and B.W.J.H.P.). This work was supported by the KNAW Academy Professor Award (PAH/6635) (to D.I.B. and used to support F.H.) and a NARSAD Young Investigator grant and by National Institute of Mental Health (NIMH) grant (R01MH125902) (H.M.vL.). *Generation Scotland:* Generation Scotland received core support from the Chief Scientist Office of the Scottish Government Health Directorates (CZD/16/6) and the Scottish Funding Council (HR03006) and is currently supported by the Wellcome Trust (216767/Z/19/Z). Genotyping of the Generation Scotland samples was carried out by the Genetics Core Laboratory at the Edinburgh Clinical Research Facility, University of Edinburgh, Scotland and was funded by the Medical Research Council UK and the Wellcome Trust (Wellcome Trust Strategic Award "STratifying Resilience and Depression Longitudinally" (STRADL) Reference 104036/Z/14/Z) (H.C.W.). This work was funded by a UKRI Medical Research Council grant for the Precision Medicine MRC DTP (MR/W006804/1) (P.Z.G.). *GLAD +:* We thank the GLAD+ Study, and NIHR BioResource volunteers for their participation and gratefully acknowledge the contributions of NIHR BioResource centres, NHS Trusts, and staff. We also thank the National Institute for Health & Care Research (NIHR), NHS Blood and Transplant, and Health Data Research UK as part of the Digital Innovation Hub Programme. This study represents independent research funded by the NIHR Biomedical Research Centre at South London and Maudsley NHS Foundation Trust and King's College London. Further information is available at (https://www.maudsleybrc.nihr.ac.uk/facilities/bioresource/). The views expressed are those of the authors and do not necessarily reflect those of the NHS, NIHR, HSC R&D Division, King's College London, or the Department of Health and Social Care. The genome-wide summary statistics for the Adams et al.[27] analysis of 23andMe, Inc., data were obtained under a data transfer agreement with QIMR Berghofer. We would like to thank the research participants and employees of 23andMe, Inc. for making this work possible.

## Author contributions

J.T.T., N.G.M. and B.L.M. conceived the study. Collection, contribution of data and/or supervision of analysts was provided by the following people for each cohort: AGDS = N.G.M., S.E.M., I.B.H., E.M.B., N.R.W., B.L.M.; QSkin = D.C.W., C.M.O.; All Of Us = E.M.D., Bionic = BIONIC consortium, D.I.B., B.W.J.H.P., H.M.vL., GLAD + = The GLAD Study, J.R.I.C., T.C.E., G.B., Generation Scotland = M.A., H.C.W. Analyses in each cohort were carried out as follows: AGDS = J.T.T., All Of Us = J.G.T., and P.Y., Bionic = F.H., GLAD + = R.W., Generation Scotland = P.Z.G., UK Biobank = J.T.T. J.T.T. performed the meta-analyses and most of the downstream statistical and bioinformatics analyses, with MiXeR analyses being carried out by J.G.T. Supervision throughout the project was provided by B.L.M. Support and advice with statistical analyses and interpretation was provided by N.R.W. and J.R.I.C. J.T.T., B.L.M. and J.G.T. wrote the manuscript, with all authors providing comments and suggestions.

## Competing interests

The authors declare no competing interests.

## Additional information

[1]Brain and Mental Health Program, QIMR Berghofer Medical Research Institute, Brisbane, QLD, Australia. [2]School of Biomedical Sciences, The University of Queensland, Brisbane, QLD, Australia. [3]Department of Biological Psychology, Faculty of Behavioral and Movement Sciences, Vrije Universiteit Amsterdam, Amsterdam, The Netherlands. [4]Department of Complex Trait Genetics, Vrije Universiteit Amsterdam, Amsterdam, The Netherlands. [5]Amsterdam Public Health Research Institute, Amsterdam, The Netherlands. [6]Division of Psychiatry, Centre for Clinical Brain Sciences, University of Edinburgh, Edinburgh, UK. [7]Social, Genetic, and Developmental Psychiatry Centre, Institute of Psychiatry, Psychology and Neuroscience, King's College London, London, UK. [8]National Institute for Health and Care Research Maudsley Biomedical Research Centre, South London and Maudsley NHS Trust, London, UK. [9]University of Queensland, Child Health Research Centre, Brisbane, QLD, Australia. [10]Brain and Mind Centre, Faculty of Medicine and Health, University of Sydney, Sydney, NSW, Australia. [11]Population Health Program, QIMR Berghofer Medical Research Institute, Brisbane, QLD, Australia. [12]Department of Psychiatry, Amsterdam UMC location Vrije Universiteit Amsterdam, Amsterdam, The Netherlands. [13]Department of Psychiatry, University of Groningen, University Medical Center Groningen, Groningen, The Netherlands. [14]Institute for Molecular Bioscience, University of Queensland, Brisbane, QLD, Australia. [15]Department of Psychiatry, University of Oxford, Oxford, UK. [16]School of Biomedical Sciences, Faculty of Health, Queensland University of Technology, Brisbane, QLD, Australia. ✉ e-mail: Jodi.Thomas@qimrb.edu.au; Brittany.Mitchell@qimrb.edu.au

## BIONIC consortium

Floris Huider ⓘ [3,4,5], Hanna M. van Loo[13], Brenda W. J. H. Penninx[5,12] & Dorret I. Boomsma ⓘ [4,5]

## The GLAD Study

Rujia Wang[7,8], Jonathan R. I. Coleman ⓘ [7,8], Thalia C. Eley ⓘ [7,8] & Gerome Breen ⓘ [7,8]

A full list of members and their affiliations appears in the Supplementary Information.

