## [Transparent Peer Review file · Nature Communications]

Sex-stratified genome-wide association meta-analysis of Major Depressive Disorder

Corresponding Author: Dr Jodi Thomas

Version 0:

Reviewer comments:

Reviewer #1

(Remarks to the Author)

In this manuscript, Thomas et al report findings from the largest sex-aware MDD GWAS meta-analyses. Their goal was to investigate whether sex differences in MDD can be explained, at least in part, by genetic effects. The sample comprised five international cohorts from Australia, the Netherlands, the UK, and the USA, all with at least 10,000 cases and matched controls.

A number of sex differences were reported. First, the sex-stratified GWAS analyses yielded 16 and eight independent genome-wide significant SNPs in females and males, respectively, though the female sample was roughly 1.7 times the size of the male sample. Second, the authors provide evidence that autosomal SNP heritability and polygenicity are both higher in females than males, suggesting that the number of SNPs contributing to MDD risk is higher in females. However, this difference is driven largely by population prevalence differences between females and males (0.2 for females, 0.1 for males). MDD may be underdiagnosed in males, however, so this could explain some of the difference. Third, the genetic correlation for MDD between females and males across the whole meta-analysis was significantly less than one, suggesting that MDD variants are not fully shared across the sexes. However, this difference did not persist after removing cohort variation by conducting a meta-analysis of the genetic correlation estimates from all six female-male within-cohort comparisons, suggesting heterogeneity between cohorts rather than a true sex difference. Fourth, the authors found a larger genetic correlation and polygenic overlap between MDD in females and BMI and metabolic syndrome than MDD in males with these same traits, suggesting that sex-specific pleiotropic effects may contribute to the higher prevalence of metabolic symptoms in females with MDD.

Overall, the paper is well-written and clear. The authors do an admirable job of identifying sex differences and investigating what could be causing them, including various factors that can affect the interpretation of the findings. For example, as noted above, some of the sex differences noted could be due to prevalence differences between females and males and heterogeneity between cohorts, rather than true sex differences.

Specific comments:

1. We suggest the authors adhere to some of the best practices outlined in the paper by Khramtsova et al. 2023 (Khramtsova, E. A., Wilson, M. A., Martin, J., Winham, S. J., He, K. Y., Davis, L. K., & Stranger, B. E. (2023). Quality control and analytic best practices for testing genetic models of sex differences in large populations. *Cell*, 186(10), 2044–2061. <https://doi.org/10.1016/j.cell.2023.04.014>). For example:
 - a. The manuscript may benefit from a brief review in the introduction stating what is meant by sex (chromosomal sex?). Also, were individuals with chromosomal aneuploidies excluded?
 - b. Rather than using the term “sex-dependent”, the authors could consider using the term “sex-differential” (as in Khramtsova et al.), as causality may not be clear.
 - c. It would also be helpful to know whether they followed the guidelines throughout their QC and analytical pipelines.
2. It would be helpful to know whether the results from different analyses align. For example, is there convergence of results from a) sex-stratified, b) genotype by sex interaction, and c) causal genomic regions identified with gwas-pw? Or do they lead to different conclusions? This should be stated and discussed more clearly.

3. Some discussion points seem contradictory. For example, from the Mixer and gwas-pw results, the authors conclude that “Both our MiXeR and gwas-pw results suggest that the set of causal variants influencing MDD in males is a subset of those that influence MDD in females.” (which is also depicted in the venn diagrams figure 2F&G), but they do find SNPs only associated with males in their sex-stratified GWAS and further down in the discussion it reads “Our sex-stratified GWAS meta-analysis of MDD has differential power and thus our male GWAS may not have had the power to identify the genes associated with MDD in females, while genes detected in males only could be male-specific”.

4. How can the different patterns of correlations in Figure 2D and E be interpreted?

5. The clarity of the manuscript may be improved by including a figure showing the sex-differential effects of identified SNPs from different analyses (e.g., as a forest plot).

6. The discussion could include a comparison of the current results to previous sex-stratified MDD GWAS results.

7. The authors might consider repeating some analyses with downsampled female datasets as a sensitivity analysis to rule out power differences. They could also perform a case-case GWAS of female MDD cases versus male MDD cases as a way to identify any opposite-effect SNPs.

8. The authors did not include any sex-stratified polygenic risk score (PRS) analyses, which could further elucidate whether the observed sex differences in genetic architecture translate into differential genetic risk profiles for MDD. For example, is predictive power greater when calculating PRS with same-sex summary statistics?

9. The authors could also consider tissue-specific gene expression analyses to assess whether certain tissues or cell types are disproportionately affected by sex-specific SNPs.

Minor comments:

1. Line 104: It would be helpful to report the estimate and confidence intervals.

2. Line 109: How are they different (opposite or different in size)?

3. Line 140-144: Consider moving the genetic correlation results with previous MDD GWAS to the section “Genetic Correlation of MDD Across Sexes”. Even though one looks at correlations with external analyses and one within the presented dataset, they try to accomplish the same. Do you see the same results?

4. Figure 2, line 199: Considering that males are likely underdiagnosed and the true population prevalences of males and females might be more similar than what is diagnosed & you do not see differences in heritabilities between males and females when assuming the same liability, wouldn't it make sense to present heritability estimates with the same assumed population prevalence here? As the difference is driven by the difference in prevalence alone, which is only an assumed and approximated concept to begin with.

5. Figure 2: In the Venn diagrams the redish/purple colour and the text contrast are too low and hard to read.

6. Line 337: Spelling of “Chromosome”

7. Line 350 & 353: Change it to “compared to MDD”

8. Figure 4: Again, the contrast between the purple background and numbers in the Venn diagram is very low. Consider a different colour or possibly write numbers in white. It also seems like a different purple compared to Figure 2.

(Remarks on code availability)

Reviewer #2

(Remarks to the Author)

(Remarks on code availability)

Reviewer #3

(Remarks to the Author)

This paper reports the results of what is unquestionably the largest sex-stratified genome wide association meta-analyses of MDD to date. The authors include several leaders in the field of psychiatric genetics and the paper reflects the considerable expertise in the analysis of genome-wide data. The study includes a number of important strengths: 1) inclusion of the X chromosome, 2) use of MDD cases based on more defined clinical (ie, DSM) criteria, 3) sample size sufficient for both sex-stratified GWAS and genotype x sex analyses and 4) a very thoughtful and detailed analyses of potential issues for sex-stratified GWAS analyses and conclusions of sex differences in genetic architecture. Finally, the paper is also timely considering the striking number of recent sex-dependent GWAS papers on a range of phenotypes including neuroanatomy (e.g., Shafee et al Nat Comm 2024; Ji et al Nat Genetics 2025). Considering the well-documented differences in the prevalence of MDD it is indeed surprising that the topic of sex-dependent molecular genetic influences on MDD has only recently been addressed. There are previously published GWASs as well as numerous analyses of twin cohorts that have

provided evidence for greater genetic influences on MDD in females, but not without conflicting results. The importance of the topic merits the stronger analysis described in this paper. In brief, this is an important contribution that includes, among several virtues, a well-constructed and accessible Results section.

The critical finding is that of a genetic correlation (r_g) that is significantly <1.0 between females and males in the sex-stratified meta-analysis. This result suggests that MDD variants are not fully shared across the sexes. Likewise, the authors report that SNP-based heritability (h^2_{SNP}) estimates were higher in females. As the authors note, this finding suggests that the amount of variation in MDD risk explained by SNPs is higher in females.

These findings go to the heart of the objective but not without important caveats. The section entitled “Genetic Correlation of MDD Across Sexes” is thus very thoughtful and important consideration of relevant issues. As noted above, the Results reveal that the genetic correlation (r_g) between females and males was significantly <1.0 . The authors considered a series of potential complications in the meta-analyses that might compromise this conclusion. Subsequent analyses did suggest that the female-male $r_g <1.0$ could reflect heterogeneity between cohorts rather than true genome-wide sex differences – with the suggestion that heterogeneity was greater across female cohorts or heterogeneity of MDD. However, removing cohort variation by conducting a meta-analysis of the correlations from all six female-male within cohort comparisons continued to reveal a correlation significantly <1.0 . This suggests sex differences in genetic effects that are not fully explained by across cohort heterogeneity. These are important analyses lacking in previous studies and inspire confidence in the conclusions.

An issue is the differences in sample size and thus power between males and females. However, the authors, obviously aware, performed a sensitivity analysis. The analyses showed the same pattern of sex differences with an actually higher h^2_{SNP} and polygenicity in females compared to males.

The authors report that in the female GWAS, SNPs were significantly enriched for genes involved in central nervous system neuron development and central nervous system neuron differentiation. This conclusion is consistent with previous analyses underscoring the importance of perinatal neural development for later psychopathology, including gene-set enrichment. I would suggest that a control analysis be included that randomly selects the same number of genes from the reference list for enrichment. Unfortunately, many databases are heavily weighted for genes involved in developmental processes (including neuronal development and neurogenesis) and our experience is that random gene lists can often produce apparent enrichment. This is a rather simple control. The authors could also strengthen this analysis if the genes of interest show increased expression in the perinatal brain by comparison to adult expression levels. Also, a rather simple analysis with Allen Brain Atlas.

The identification of a genomic region containing a variant that mapped to the FTO gene is particularly interesting. Does this suggest that co-morbidities in females between MDD and consummatory behavior might emerge as a function of a common genetic basis? This gene seems particularly interesting considering the female-specific link to cardiometabolic GWAS findings. As an aside, this sex-dependent genetic correlation was also reported by Silveira et al and could be noted.

The authors employed both sex stratification and genotype x sex interaction approaches. Can they comment on the merits of the two approaches?

Finally, I would ask the reviewers to comment, however briefly, on the rather obvious implications for novel therapeutics.

(Remarks on code availability)

Version 1:

Reviewer comments:

Reviewer #1

(Remarks to the Author)

The authors have done an excellent job responding to the critique, performing additional analyses, and updating the manuscript. I have no further concerns and think this paper makes a valuable contribution to the field.

(Remarks on code availability)

Reviewer #2

(Remarks to the Author)

(Remarks on code availability)

Reviewer #3

(Remarks to the Author)

The authors have done an admirable job of responding constructively to all comments. I have no further concerns. This is an important contribution to the literature.

(Remarks on code availability)

Reviewer #1 (Remarks to the Author):

In this manuscript, Thomas et al report findings from the largest sex-aware MDD GWAS meta-analyses. Their goal was to investigate whether sex differences in MDD can be explained, at least in part, by genetic effects. The sample comprised five international cohorts from Australia, the Netherlands, the UK, and the USA, all with at least 10,000 cases and matched controls.

A number of sex differences were reported. First, the sex-stratified GWAS analyses yielded 16 and eight independent genome-wide significant SNPs in females and males, respectively, though the female sample was roughly 1.7 times the size of the male sample. Second, the authors provide evidence that autosomal SNP heritability and polygenicity are both higher in females than males, suggesting that the number of SNPs contributing to MDD risk is higher in females. However, this difference is driven largely by population prevalence differences between females and males (0.2 for females, 0.1 for males). MDD may be underdiagnosed in males, however, so this could explain some of the difference. Third, the genetic correlation for MDD between females and males across the whole meta-analysis was significantly less than one, suggesting that MDD variants are not fully shared across the sexes. However, this difference did not persist after removing cohort variation by conducting a meta-analysis of the genetic correlation estimates from all six female-male within-cohort comparisons, suggesting heterogeneity between cohorts rather than a true sex difference. Fourth, the authors found a larger genetic correlation and polygenic overlap between MDD in females and BMI and metabolic syndrome than MDD in males with these same traits, suggesting that sex-specific pleiotropic effects may contribute to the higher prevalence of metabolic symptoms in females with MDD.

Overall, the paper is well-written and clear. The authors do an admirable job of identifying sex differences and investigating what could be causing them, including various factors that can affect the interpretation of the findings. For example, as noted above, some of the sex differences noted could be due to prevalence differences between females and males and heterogeneity between cohorts, rather than true sex differences.

Dear Reviewer 1: Thank you for your time reviewing our manuscript and your positive comments about our study. Your suggestions were highly valuable and have enhanced the quality of this research. Please see our responses in red below each point. Line numbers refer to the clean version/track changes version with 'No markup' shown.

Specific comments:

1. We suggest the authors adhere to some of the best practices outlined in the paper by Khramtsova et al. 2023 (Khramtsova, E. A., Wilson, M. A., Martin, J., Winham, S. J., He, K. Y., Davis, L. K., & Stranger, B. E. (2023). Quality control and analytic best practices for testing genetic models of sex differences in large populations. *Cell*, 186(10), 2044–2061. <https://doi.org/10.1016/j.cell.2023.04.014>).

For example:

a. The manuscript may benefit from a brief review in the introduction stating what is meant by sex (chromosomal sex?). Also, were individuals with chromosomal aneuploidies excluded?

Thank you for this comment, we agree this is important to clarify. Throughout the manuscript sex is used as recommended in Khramtsova et al., 2023 and the Nature Portfolio Research Ethics guidelines, i.e. in reference to biological differences between females and males. When writing this manuscript we took care to only use the terms sex/female/male and to not use the terms gender/woman/man so that the different concepts of sex and gender are not confused. To clarify this, we have added a sentence right at the start of the introduction:

L59: 'Throughout this manuscript, the term 'sex' refers to differences in biological characteristics between females and males.'

In terms of our research, sex was defined as the chromosomal composition of individuals and sex chromosome aneuploidies were not included in this research. To clarify this, we have included statements about how sex was defined in our study:

Results L134: 'We conducted genome-wide association studies (GWAS) of Major Depressive Disorder (MDD) in five new cohorts. These were meta-analysed with previously published GWAS meta-analysis summary statistics from Blokland et al. [16] for each sex separately. Sex was defined by chromosomal composition, with XX individuals as female and XY as male.'

Methods L599: 'Sex was defined by chromosomal composition, with XX individuals classified as female and XY as male'

b. Rather than using the term “sex-dependent”, the authors could consider using the term “sex-differential” (as in Khramtsova et al.), as causality may not be clear.

As in Khramtsova et al., sex-differential is defined as SNPs which have an effect in both sexes but the effect size is larger in one sex compared to the other. Sex-opposite is defined as SNPs having opposite effects in each sex. In our manuscript, sex-dependent refers to both sex-differential and sex-opposite. To clarify this definition, we have revised the following text:

Introduction L92: 'Sex-dependent effects refer to genetic variants that affect both sexes, but with effect sizes that differ in magnitude or direction between females and males. Sex-specific effects suggest that different genetic variants may contribute to MDD in males and females.'

Discussion L429: '1) sex-dependent effects in which genetic variants for MDD have differing magnitudes and/or direction of effects across sexes'

c. It would also be helpful to know whether they followed the guidelines throughout their QC and analytical pipelines.

Thank you for bringing our attention to this publication regarding best practices for sex-aware analyses. We have created a supplementary table collating the main best practices outlined in the publication by Khramtsova et al., 2023 and indicated whether each of these were addressed in our 6 cohorts (5 cohorts included in the meta-analysis and 1 cohort used as replication). We have also included a reference to this in the Methods main text:

L602: 'Table S33 summarises how the recommended best practices for sex-aware analyses from Khramtsova et al. [39] were addressed.'

As the analyses in this manuscript relied on using genotype data that had previously been QCed and imputed (our Supplementary Methods A refer to the appropriate publications for each cohort), most of the guidelines for sex-aware QC were unfortunately not followed in our cohorts. Therefore, the addition of this information in the added supplementary table provides transparent reporting of this information. To further address this issue, we have also added a sentence in the limitations section of our discussion:

L570: 'Lastly, quality control of the genotype data was performed prior to this study and, as such, most recommended sex-aware quality control practices [39] were not implemented, potentially introducing undetected sex-specific technical biases.'

2. It would be helpful to know whether the results from different analyses align. For example, is there convergence of results from a) sex-stratified, b) genotype by sex interaction, and c) causal genomic regions identified with gwas-pw? Or do they lead to different conclusions? This should be stated and discussed more clearly.

Thank you for this suggestion. We have added text and supplementary figures to better compare the results across our different analyses and feel that the alignment of our results across the different methods is now much clearer.

In the results section, we have added text that compares the results between our sex-stratified GWAS and our MiXeR and gwas-pw analyses:

L302: 'All 42 shared regions identified by gwas-pw exhibit peaks of MDD association in both sexes (Figure S11). Furthermore, the four possible causal risk variants within these shared regions had concordant effect directions across sexes (Figure S12). Of the three female-specific regions (gwas-pw), two contain genome-wide significant SNPs in females only and the third includes a SNP nearing significance in females only (Figure S11). This suggests some genome-wide significant SNPs identified in females may be female-specific, not only due to lower male GWAS power. Neither MiXeR nor gwas-pw identified male-specific causal variants/regions. All but one genome-wide significant autosomal SNP in males fell within shared regions (gwas-pw). Thus, SNPs reaching significance in males only may result from stronger effects in males rather than male-specificity, with weaker associations in females going undetected due to power limitations. As gwas-pw and MiXeR only analysed autosomes, male-specific variants may exist on the X chromosome, where one SNP was genome-wide significant in males only.'

This refers to the original supplementary Figure 11, which we presented previously and displays the sex-stratified GWAS Miami plot overlaid with the regions identified by gwas-pw. It also refers to Supplementary Figure 12, which is newly made based on these suggestions and presents forest plots showing effect sizes of the lead genome-wide significant SNPs identified in the female GWAS and male GWAS, lead nominally significant SNPs identified in the GxS analysis and the possible causal variants for MDD identified by gwas-pw as shared by both sexes (also addresses comment 5 below).

We have also added text in our Supplementary Results C comparing the results from our GxS analysis to our sex-stratified GWAS and gwas-pw analysis:

'None of the four SNPs identified as nominally significant in the genotype-by-sex interaction analysis (GxS) reached genome-wide significant in either the female or male GWAS (Figure S12). Two of the four nominally significant independent SNPs from the GxS analysis showed opposite effects by sex,

with the minor allele increasing MDD risk in females and decreasing risk in males. These SNPs are located on chromosome 1 (rs12092435, $p = 4.41 \times 10^{-7}$) and chromosome 19 (rs28573687, $p = 6.57 \times 10^{-7}$). The SNP on chromosome 12 (rs12312238, $p = 7.7 \times 10^{-7}$) has no effect in females but decreases MDD risk in males, while the SNP on chromosome 20 (rs6080675, $p = 6.24 \times 10^{-8}$) increases the risk of MDD in both sexes with a larger effect in males (Figure S12D).

Consistent with our gwas-pw results, none of the causal regions identified by gwas-pw as shared between sexes exhibited a peak of association in the GxS analysis (Figure S22). However, none of the female-specific regions identified by gwas-pw exhibited a peak of association in the GxS analyses (Figure S22). This was an unexpected finding as we would expect sex-specific regions to be identified in the GxS analysis. This discrepancy may be due to limited statistical power, as genome-wide interaction analyses require very large sample sizes to reliably detect such effects.'

This refers to Supplementary Figure 22, a new figure which displays the GxS Manhattan plot overlaid with the regions identified by gwas-pw.

3. Some discussion points seem contradictory. For example, from the Mixer and gwas-pw results, the authors conclude that "Both our MiXeR and gwas-pw results suggest that the set of causal variants influencing MDD in males is a subset of those that influence MDD in females." (which is also depicted in the venn diagrams figure 2F&G), but they do find SNPs only associated with males in their sex-stratified GWAS and further down in the discussion it reads "Our sex-stratified GWAS meta-analysis of MDD has differential power and thus our male GWAS may not have had the power to identify the genes associated with MDD in females, while genes detected in males only could be male-specific".

We agree that some discussion points seemed contradictory. We have revised the text to address this and clarify these results.

Results L308: 'Neither MiXeR nor gwas-pw identified male-specific causal variants/regions. All but one genome-wide significant autosomal SNP in males fell within shared regions (gwas-pw). Thus, SNPs reaching significance in males only may result from stronger effects in males rather than male-specificity, with weaker associations in females going undetected due to power limitations. As gwas-pw and MiXeR only analysed autosomes, male-specific variants may exist on the X chromosome, where one SNP was genome-wide significant in males only.'

We have also rephrased the section in the discussion that you mention and feel that is now clearer:

L517: 'Due to the differential power of our sex-stratified GWAS meta-analysis, it is difficult to determine whether any of the observed gene associations reflect sex-dependent or sex-specific effects. Our polygenic overlap results suggest the presence of shared and female-specific causal variants, but no male-specific causal variants. The genes identified only in females may be female-specific, but this should be interpreted with caution, as the smaller male GWAS sample may have lacked the power to detect these SNPs identified in females. Conversely, the genes identified only in males may not be male-specific, but rather sex-dependent with stronger effects in males. This could explain why they were detected despite the lower power of the male GWAS.'

4. How can the different patterns of correlations in Figure 2D and E be interpreted?

Thank you for highlighting this, we had explained the patterns within each of the r_g (Figure 2D) and correlation of effect sizes (Figure 2E), but not across these two analyses. To address this, we have added a paragraph below:

L274: 'R_g captures the shared genetic architecture across the entire genome whereas the Pearson correlation of effect sizes focuses only on the lead genome-wide significant SNPs previously associated with MDD in a sex-combined GWAS meta-analysis [27]. This distinction may explain why female-male within cohort comparisons showed an r_g estimate not significantly different from one (Figure 2D), but a correlation significantly less than one (Figure 2E). The female-male r_g (using our sex-stratified meta-analysis results) being significantly less than one could be driven by between cohort heterogeneity, whereas specific lead SNPs may still exhibit sex-dependent differences in effect sizes.'

5. The clarity of the manuscript may be improved by including a figure showing the sex-differential effects of identified SNPs from different analyses (e.g., as a forest plot).

Thank you for this idea. Based on your suggestion, we have added Supplementary Figure 12 which shows the effect sizes from the female and male GWAS, of SNPs reaching genome-wide significance in the female GWAS, SNPs reaching genome-wide significance in the male GWAS, SNPs reaching nominal significance in the GxS analysis, and possible causal variants for MDD shared by both sexes as identified by gwas-pw (no possible causal risk variants were identified within the three female-specific genomic regions identified by gwas-pw).

6. The discussion could include a comparison of the current results to previous sex-stratified MDD GWAS results.

We appreciate the reviewer's comment and note that there are two prior sex-stratified MDD GWAS publications (Blokland et al. [citation 16] and Silveira et al. [citation 17]). Noting, that the Blokland results have been incorporated into our meta-analysis and the Silveira results only included UK Biobank - which we have also included in our meta-analysis. Below, we highlight where we have already compared our findings to these studies in the Discussion section:

L447: 'Autosomal SNP-based heritability (h^2_{SNP}) was higher in females than males, and remained consistent in our sensitivity analyses accounting for the differential power across sexes and for across-cohort heterogeneity. This result reflects previous findings in a sex-stratified GWAS [16]'

(note we have not compared to Silveira as this publication did not calculate heritability)

L495: '... A genetic correlation not significantly different to one was found by Blokland et al. [16] but not by Silveira et al. [17].....'

To expand on this, we have also added new text to the discussion comparing our results to these previous sex-stratified MDD GWAS results:

L530: 'We observed stronger genetic correlations and greater polygenic overlap between MDD in females and metabolic traits (body mass index and metabolic syndrome) than MDD in males with these same traits. Similarly, Silveira et al. [17] found genetic correlations between depression and metabolic features (including body mass index, waist-to-hip ratio and triglycerides), as well as coronary artery disease, were larger and significant only in females.'

L544: 'We found a significantly higher genetic correlation between regular smoking and MDD in females than in males, consistent with Silveira et al. [17].'

7. The authors might consider repeating some analyses with downsampled female datasets as a sensitivity analysis to rule out power differences. They could also perform a case-case GWAS of female MDD cases versus male MDD cases as a way to identify any opposite-effect SNPs.

Yes, we definitely agree that the differential power of our female and male GWAS is a limitation and we have mentioned this in the discussion limitations. Furthermore, we have already included a downsampling of the female dataset as you suggest. However, your comment suggests that this was not clear enough so we have reworded the following text to improve the clarity around what sensitivity analyses we did:

L183: 'We conducted a range of sensitivity analyses. For h^2_{SNP} , we used LDSC as an alternative method and explored the impact of lifetime population prevalence and unscreened controls (given evidence that MDD could be under-diagnosed in males [29]). For h^2_{SNP} , π and S , we 1) assessed the effect of differential power between female and male GWAS by repeating analyses using equivalent sample sizes in females and males, and 2) accounted for across-cohort heterogeneity [25].'

L284: 'Bivariate MiXeR [29] revealed that all 7,111 ($SE = 701$) causal variants for MDD in males were shared with MDD in females, with an additional 6,133 ($SE = 988$) variants unique to MDD in females and zero ($SE = 0.0004$) variants unique to MDD in males (Figure 2F). For those causal variants shared by females and males, the correlation of their effect sizes was 1.0 ($SE = 6.4 \times 10^{-8}$) (Table S11). We also accounted for the differential power across our sex-stratified GWAS by repeating MiXeR analyses using equivalent sample sizes in females and males and found the same pattern (Table S12, Figure S31).'

Thank you for your suggestion to perform a case-case GWAS to identify any potential SNPs with opposite effects between sexes. We have decided not to include this analysis, as we believe it would not provide additional insights beyond those already captured in our current results. The MiXeR analyses we conducted are capable of detecting SNPs with opposite effects across sexes. Our results from the MiXeR analyses showed that for causal variants shared between females and males, the correlation of effect sizes was 1.0 ($SE = 6.4 \times 10^{-8}$) (Table S11). This suggests that all of the SNPs shared between sexes have an effect size in the same direction. We also repeated this analysis with equivalent sample sizes in females and males and found the same results (Table S12).

8. The authors did not include any sex-stratified polygenic risk score (PRS) analyses, which could further elucidate whether the observed sex differences in genetic architecture translate into differential genetic risk profiles for MDD. For example, is predictive power greater when calculating PRS with same-sex summary statistics?

Thank you for this suggestion. We had considered including sex-stratified PRS analyses but ultimately decided not to for several reasons. First, at QIMR Berghofer we have limited access to independent cohorts not already included in the meta-analysis that we could use to conduct these PRS analyses. Second, PRS prediction accuracy is highly dependent on the power of the discovery GWAS, and our female and male GWAS meta-analyses differ substantially in their power. This disparity makes direct comparisons of PRS performance between sexes difficult to interpret without downsampling, which would require reducing the effective sample size of the full female meta-analysis. Unfortunately, downsampling is not feasible in our case due to the inclusion of multiple cohorts as well as previously published sex-stratified summary statistics, which also differ in power. For these reasons, we have decided not to include sex-stratified PRS analyses in the current

manuscript. However, we agree this is an important direction for future work and have acknowledged this in the discussion section:

L507: 'Polygenic risk score (PRS) analyses may provide another avenue for future research to assess whether the genetic architecture of MDD differs between sexes. However, it will be important to account for the differential power of sex-stratified GWAS used to construct PRS [39].'

9. The authors could also consider tissue-specific gene expression analyses to assess whether certain tissues or cell types are disproportionately affected by sex-specific SNPs.

We agree that tissue-specific gene expression analyses are an interesting avenue with which to explore our sex-stratified GWAS results. Indeed, we did include this analysis in the manuscript. Please see the results section '*Functional annotation and analyses*' in which we have slightly updated the text for clarity:

L339. 'Gene-property analysis for tissue specificity using 30 general tissue types (GTEx v8) identified SNPs from both the female and male stratified GWAS analyses as significantly enriched for gene expression in brain tissue, while pituitary tissue was significantly enriched only in the female SNPs (Figure S15A). Furthermore, using 53 tissues types (GTEx v8), female and male SNPs were significantly enriched for gene expression in the cortex and frontal cortex, caudate, putamen and nucleus accumbens basal ganglia, hippocampus, amygdala, hypothalamus and anterior cingulate cortex. Female, but not male, SNPs were significantly enriched for gene expression in the cerebellum and cerebellar hemisphere (Figure S15B). It is likely the differential power across the female and male GWAS meta-analysis results led to the difference in significant results for the gene-set and tissue analyses.'

Please also see Supplementary Results C and Figure S19 where we present the results of a tissue-specific gene expression analysis using the GxS summary statistics:

'We found a significantly higher genetic correlation between regular smoking and MDD in females than in males, consistent with Silveira et al. [17].'

We also updated the text in the methods section '*Functional annotation and analyses*' to increase the clarity around this analysis and what it determines:

L777: 'Gene property analysis for tissue specificity was conducted to determine whether genes that are more highly expressed in specific tissues tend to be more associated with depression. We used RNA-sequencing data from 30 general tissue types and 53 tissue types from GTEx v8 [86].'

To address your suggestion further, we have also added another tissue-specific gene expression analysis determining whether genes that are more highly expressed in different developmental stages of the brain are more associated with depression in each sex and from the GxS analysis. This analysis has been added in the Results:

L334: 'To explore further, we conducted a gene-property analysis using BrainSpan gene expression data across 11 brain developmental stages. SNPs from both sexes were significantly enriched for expression in the 'late mid-prenatal' stage, with additional 'early mid-prenatal' enrichment in females (Figure S14). These findings support the involvement of neurodevelopmental processes indicated by gene-set analysis.'

The Results from this analysis using the GxS summary statistics can also be found in Supplementary Results C:

'Furthermore, SNPs from the GxS interaction analysis were not enriched for gene expression in any of the 11 brain developmental stages (Figure S21).'

We have also added Supplementary Figures 14 and 21 which display the results from the gene property analysis using 11 general developmental stages of brain samples (BrainSpan) carried out using the sex-stratified and GxS summary statistics.

We have also included details on this analysis in the Methods:

L777: 'Gene property analysis for tissue specificity was conducted to determine whether genes that are more highly expressed in specific tissues tend to be more associated with depression. We used RNA-sequencing data from 30 general tissue types and 53 tissue types from GTEx v8 [86], as well as from 11 general developmental stages of brain samples from BrainSpan [96].'

Minor comments:

1. Line 104: It would be helpful to report the estimate and confidence intervals.

This has been added:

L105: 'For instance, a study using data from the UK Biobank reported a SNP-based genetic correlation between broad depression in males and females that was significantly less than one ($r_g = 0.91$, standard error (SE) not reported) [17], however this was not the case in a large sex-stratified meta-analysis of MDD conducted by the Psychiatric Genomics Consortium ($r_g = 1.01$, SE = 0.2) [16].'

2. Line 109: How are they different (opposite or different in size)?

Opposite – the text has been updated to reflect this:

L111: 'Furthermore, a cross-disorder (schizophrenia, bipolar disorder, MDD) genotype-by-sex interaction analysis identified a locus with opposite effects in females and males, further pointing to the role of sex-dependent genetic factors [16].'

3. Line 140-144: Consider moving the genetic correlation results with previous MDD GWAS to the section "Genetic Correlation of MDD Across Sexes". Even though one looks at correlations with external analyses and one within the presented dataset, they try to accomplish the same. Do you see the same results?

We have moved these results as suggested. The relevant Results section in 'Sex-stratified GWAS' now only compares our results to the most recent sex-combined GWAS:

L144: 'Our sex-stratified results showed high genetic correlation with the largest GWAS meta-analysis of MDD in both sexes combined [27] (Female-both sexes $r_g = 0.98 \pm 0.01$, Male-both sexes $r_g = 0.92 \pm 0.02$) (Figure S1, Tables S4-S5).'

And the Results section 'Genetic Correlation of MDD Across Sexes' now contains the information comparing our sex-stratified GWAS to previous sex-stratified GWAS:

L225: 'There was a significantly higher r_g between the previous female GWAS of MDD by Blokland et al. [16] and our female GWAS than with our male GWAS ($Z = 4.49$, $\text{padj}(\text{Benjamini-Hochberg}) = 7.3 \times 10^{-6}$). The male GWAS by Blokland et al. [16] showed a significantly higher r_g with our male GWAS than with our female GWAS ($Z = 3.23$, $\text{padj}(B-H) = 0.003$). A similar, though non-significant, trend was observed when comparing our sex-stratified GWAS to those from Silveira et al. [17] (Figure S1, Tables S4-S5).'

4. Figure 2, line 199: Considering that males are likely underdiagnosed and the true population prevalences of males and females might be more similar than what is diagnosed & you do not see differences in heritabilities between males and females when assuming the same liability, wouldn't it make sense to present heritability estimates with the same assumed population prevalence here? As the difference is driven by the difference in prevalence alone, which is only an assumed and approximated concept to begin with.

We appreciate the reviewer's thoughtful comment and agree that underdiagnosis in males could mean that the true population prevalence of MDD is more similar between sexes than current diagnostic rates suggest. In our study, however, most MDD cases were identified using DSM-5 criteria via participant-completed questionnaires, including structured tools such as the CIDI-SF (see Supplementary Table 1). This design may reduce some of the diagnostic bias typically seen in clinical settings due to factors like lower help-seeking among males.

To address the potential impact of underdiagnosis, we conducted two sensitivity analyses:

1. **Varying prevalence assumptions:** We estimated SNP-based heritability (h^2) on the liability scale using a range of population prevalence values for males and females (Supplementary Figure 3G).
2. **Accounting for unscreened controls in males:** We estimated male h^2 on the liability scale accounting for unscreened controls and a corresponding increase in population prevalence (Supplementary Figure 3H).

The results from these sensitivity analyses are presented in detail in *Supplementary Results A: Male Under-diagnosis*.

In the main text, we chose to present h^2 estimates using prevalences of 0.2 for females and 0.1 for males as these are the most widely reported previous estimates. This decision enables easier comparison with previous findings in depression genetics. To ensure transparency, we state the assumed prevalence values in the main text where relevant, including the Figure 2 legend.

To guide readers who may not consult the supplementary materials, we reference the sensitivity analyses directly in the main text. For example, in the Results:

L190: 'When the same population prevalence was used for males and females, we found that h^2_{SNP} on the liability scale is similar across sexes, suggesting the sex difference in h^2_{SNP} is driven by the prevalence difference (Figure S3G).'

And in the Discussion:

L452: 'We found that the sex difference in h^2_{SNP} may be driven by the prevalence difference, as when the same population prevalence was used across sexes h^2_{SNP} on the liability scale was similar in females and males. It is possible that MDD must be more severe in males compared with females for an individual to cross the liability threshold and for MDD to manifest. This increased male severity

could explain the lower prevalence of MDD in males. As more severe phenotypes tend to have a higher heritability [36], it could also explain the disappearance of the sex difference when h^2_{SNP} was calculated on the liability scale assuming equal population prevalence across sexes. Another explanation for the lower prevalence in males could be due to under-reporting and under-diagnosis of MDD in males [7, 37]. If this is the case, the true male population prevalence may be higher than reported and controls may include unidentified cases, resulting in an underestimation of male h^2_{SNP} [38]. However, in our analyses h^2_{SNP} remained higher in females even when accounting for up to 30% unscreened male controls, and a corresponding increase in male population prevalence, when compared to females with a population prevalence of 20% and no unscreened controls. Thus, h^2_{SNP} could be higher in females even with under-reporting and under-diagnosis in males, although further research is needed to quantify this under-diagnosis.'

Finally, we note that the higher polygenicity of MDD observed in females (a measure not influenced by prevalence assumptions) supports the possibility of genuine sex-specific genetic architecture, which may extend beyond diagnostic biases.

5. Figure 2: In the Venn diagrams the redish/purple colour and the text contrast are too low and hard to read.

Thank you for pointing this out, figure 2 has been updated so the numbers within the Venn diagram are in bold and the numbers in the purple circle are in white to improve the contrast and readability.

6. Line 337: Spelling of "Chromosome"

Corrected thank you (now L365 – Figure 3 legend).

7. Line 350 & 353: Change it to "compared to MDD"

Corrected (now L380 and L384).

8. Figure 4: Again, the contrast between the purple background and numbers in the Venn diagram is very low. Consider a different colour or possibly write numbers in white. It also seems like a different purple compared to Figure 2.

Figure 4 has been updated so the numbers within the Venn diagram are in bold and the numbers in the purple circles are now in white as suggested. Good spotting with the colours thankyou – they were actually the same colour but with a slightly different alpha level (transparency). This has been corrected to be the same for Figures 2 and 4.

Reviewer #2 (Remarks to the Author):

Thank you for spending the time to co-review our manuscript!

Reviewer #3 (Remarks to the Author):

This paper reports the results of what is unquestionably the largest sex-stratified genome wide association meta-analyses of MDD to date. The authors include several leaders in the field of psychiatric genetics and the paper reflects the considerable expertise in the analysis of genome-wide data. The study includes a number of important strengths: 1) inclusion of the X chromosome, 2) use of MDD cases based on more defined clinical (ie, DSM) criteria, 3) sample size sufficient for both sex-stratified GWAS and genotype x sex analyses and 4) a very thoughtful and detailed analyses of potential issues for sex-stratified GWAS analyses and conclusions of sex differences in genetic architecture. Finally, the paper is also timely considering the striking number of recent sex-dependent GWAS papers on a range of phenotypes including neuroanatomy (e.g., Shafee et al Nat Comm 2024; Ji et al Nat Genetics 2025). Considering the well-documented differences in the prevalence of MDD it is indeed surprising that the topic of sex-dependent molecular genetic influences on MDD has only recently been addressed. There are previously published GWASs as well as numerous analyses of twin cohorts that have provided evidence for greater genetic influences on MDD in females, but not without conflicting results. The importance of the topic merits the stronger analysis described in this paper. In brief, this is an important contribution that includes, among several virtues, a well-constructed and accessible Results section.

The critical finding is that of a genetic correlation (r_g) that is significantly <1.0 between females and males in the sex-stratified meta-analysis. This result suggests that MDD variants are not fully shared across the sexes. Likewise, the authors report that SNP-based heritability (h^2_{SNP}) estimates were higher in females. As the authors note, this finding suggests that the amount of variation in MDD risk explained by SNPs is higher in females.

These findings go to the heart of the objective but not without important caveats. The section entitled "Genetic Correlation of MDD Across Sexes" is thus very thoughtful and important consideration of relevant issues. As noted above, the Results reveal that the genetic correlation (r_g) between females and males was significantly <1.0 . The authors considered a series of potential complications in the meta-analyses that might compromise this conclusion. Subsequent analyses did suggest that the female-male $r_g <1.0$ could reflect heterogeneity between cohorts rather than true genome-wide sex differences – with the suggestion that heterogeneity was greater across female cohorts or heterogeneity of MDD. However, removing cohort variation by conducting a meta-analysis of the correlations from all six female-male within cohort comparisons continued to reveal a correlation significantly <1.0 . This suggests sex differences in genetic effects that are not fully explained by across cohort heterogeneity. These are important analyses lacking in previous studies and inspire confidence in the conclusions.

Thank you very much for your time reviewing our manuscript, your positive comments about our study and your thoughtful suggestions on how to improve. We were very pleased to read that you felt this study is an important and timely contribution to the field of depression genetics. We have addressed each of your comments below in red text. Line numbers refer to the clean version/track changes version with 'No markup' shown.

An issue is the differences in sample size and thus power between males and females. However, the authors, obviously aware, performed a sensitivity analysis. The analyses showed the same pattern of sex differences with an actually higher h^2_{SNP} and polygenicity in females compared to males. The authors report that in the female GWAS, SNPs were significantly enriched for genes involved in

central nervous system neuron development and central nervous system neuron differentiation. This conclusion is consistent with previous analyses underscoring the importance of perinatal neural development for later psychopathology, including gene-set enrichment. I would suggest that a control analysis be included that randomly selects the same number of genes from the reference list for enrichment. Unfortunately, many databases are heavily weighted for genes involved in developmental processes (including neuronal development and neurogenesis) and our experience is that random gene lists can often produce apparent enrichment. This is a rather simple control.

Thank you for this suggestion about a potential bias here. We would like to clarify that we did not carry out an overrepresentation analysis based on a predefined list of significant genes (e.g. the 16 genome-wide significant genes from the female GWAS). Rather our gene set enrichment analysis was conducted using the MAGMA framework implemented in FUMA SNP2GENE, which uses the gene-level association statistics of all genes (from the MAGMA gene-based test). As a competitive gene-set analysis, it tests whether genes in a given gene set (e.g., GO Terms) are more strongly associated with depression than the remaining genes not in this gene set, while accounting for potential confounding factors such as gene size, gene density, and linkage disequilibrium. Because this analysis evaluates enrichment in the context of the entire genome, a control analysis involving random gene selection from the reference list would not be applicable as a control analysis.

We recognise that our manuscript did not make this methodology sufficiently clear. Therefore, we have updated the text to clarify this analysis:

Results L329: 'Gene-set analysis using gene-level statistics for all genes revealed that...'

Methods L771: 'The gene-set analysis used the gene-level association statistics for all genes from the MAGMA gene-based test and 10,678 gene sets (4,761 curated gene sets and 5,917 Gene Ontology terms) from MsigDB v6.2 [95] with significance determined using Bonferroni corrected P-values. A competitive gene-set analysis was used which tests whether genes in a given gene set are more strongly associated with MDD than the remaining genes not in this gene set, and potential confounding factors such as gene size, gene density and linkage disequilibrium are accounted for.'

To further address your concern we have also included text about this limitation of the analysis:

Results L333: 'As many gene-set databases emphasise developmental genes, the female enrichments may partly reflect this bias.'

In addition, to provide a second test of the enrichment of gene expression across developmental periods we conducted a gene-property analysis using developmental stage-specific gene expression data from BrainSpan (which also addresses your comment below). This analysis found complementary evidence for the role of neurodevelopmental processes.

Results L334: 'To explore further, we conducted a gene-property analysis using BrainSpan gene expression data across 11 brain developmental stages. SNPs from both sexes were significantly enriched for expression in the 'late mid-prenatal' stage, with additional 'early mid-prenatal' enrichment in females (Figure S14). These findings support the involvement of neurodevelopmental processes indicated by gene-set analysis.'

The authors could also strengthen this analysis if the genes of interest show increased expression in the perinatal brain by comparison to adult expression levels. Also, a rather simple analysis with Allen Brain Atlas.

Thank you for this suggestion. We have added an analysis in which we determine whether genes that are more highly expressed in different developmental stages of the brain are more strongly associated with depression (run separately using the female GWAS summary statistics, male GWAS summary statistics and GxS summary statistics). For this analysis, we used the RNA-sequencing data from BrainSpan: Atlas of the Developing Human Brain. This analysis has been added in the Results:

L334: 'To explore further, we conducted a gene-property analysis using BrainSpan gene expression data across 11 brain developmental stages. SNPs from both sexes were significantly enriched for expression in the 'late mid-prenatal' stage, with additional 'early mid-prenatal' enrichment in females (Figure S14). These findings support the involvement of neurodevelopmental processes indicated by gene-set analysis.'

The Results from this analysis using the GxS summary statistics can also be found in Supplementary Results C:

'Furthermore, SNPs from the GxS interaction analysis were not enriched for gene expression in any of the 11 brain developmental stages (Figure S21).'

We have also added Supplementary Figures 14 and 21 which display the results from the gene property analysis using 11 general developmental stages of brain samples (BrainSpan) carried out using the sex-stratified and GxS summary statistics.

Details of this analysis have been added in the Methods:

L777: 'Gene property analysis for tissue specificity was conducted to determine whether genes that are more highly expressed in specific tissues tend to be more associated with depression. We used RNA-sequencing data from 30 general tissue types and 53 tissue types from GTEx v8 [86], as well as from brain samples collected at 11 general developmental stages from BrainSpan [96].'

The identification of a genomic region containing a variant that mapped to the FTO gene is particularly interesting. Does this suggest that co-morbidities in females between MDD and consummatory behavior might emerge as a function of a common genetic basis? This gene seems particularly interesting considering the female-specific link to cardiometabolic GWAS findings. As an aside, this sex-dependent genetic correlation was also reported by Silveira et al and could be noted.

While definitely an interesting association, the FTO gene was found to be shared by MDD and BMI as well as MDD and Metabolic Syndrome in both sexes. Therefore, it appears that the co-morbidity between MDD and metabolic traits may emerge as a function of a common genetic basis that includes the FTO gene. However, this is shared by both sexes and is not sex-specific. The relevant text from the original Results section is copied below:

L398: 'In both sexes, we identified one genomic region containing a causal variant for both MDD and BMI, which mapped to the genes FTO and IRX3 (Table S28).'

L403: 'For MDD and metabolic syndrome, we identified four genomic regions containing a causal variant for both traits shared by both sexes, which also mapped to the FTO gene (Table S30).'

Thank you for your comment regarding the similarity in our rg results and those of Silveira et al. We have added this comparison by adding new text in our Discussion:

L530: 'We observed stronger genetic correlations and greater polygenic overlap between MDD in females and metabolic traits (body mass index and metabolic syndrome) than MDD in males with these same traits. Similarly, Silveira et al. [17] found genetic correlations between depression and metabolic features (including body mass index, waist-to-hip ratio and triglycerides), as well as coronary artery disease, were larger and significant only in females

L544: 'We found a significantly higher genetic correlation between regular smoking and MDD in females than in males, consistent with Silveira et al. [17].'

The authors employed both sex stratification and genotype x sex interaction approaches. Can they comment on the merits of the two approaches?

Thank you for this suggestion. We have now included text in our Results section commenting on these two approaches and the different information they provide:

L351: 'The sex-stratified GWAS meta-analysis estimated the effect size and direction of SNP associations with MDD in each sex separately. To complement these analyses, we conducted a genome-wide genotype-by-sex interaction (GxS) meta-analysis, which directly tested whether SNP associations with MDD differ significantly between sexes. This GxS meta-analysis combined results from the same five cohorts with those from Blokland et al. [16].'

We have also included more information in the Methods:

L624: 'The sex-stratified GWAS was used to characterise the SNP effect sizes and directions of effect in each sex separately. These summary statistics were also leveraged for downstream analyses, such as SNP-based heritability, genetic correlation and polygenic overlap across sexes. The GxS analysis complements the sex-stratified GWAS by formally testing whether the differences in SNP effect sizes between sexes are statistically different.'

Finally, I would ask the reviewers to comment, however briefly, on the rather obvious implications for novel therapeutics.

We have added text at the end of the Introduction to address this comment:

L127: 'These insights into sex-specific genetic mechanisms not only deepen our understanding of the aetiology of MDD but may also inform the development of novel therapeutics that are tailored to sex-specific genetic risk profiles, ultimately contributing to more targeted and effective precision medicine strategies for MDD.'

As well as at the end of the Discussion:

L581: 'Moving forward, integrating sex-specific genetic findings into clinical practice could pave the way for more personalized diagnostic and therapeutic strategies for MDD. For example, these results may inform the development of novel therapeutics that target sex-dependent biological pathways, ultimately improving treatment efficacy and outcomes.'